



# Fatalities associated with the weather in the Czech Republic, 2000–2019

Rudolf Brázdil[1,2], Kateřina Chromá[2], Lukáš Dolák[1,2], Jan Řehoř[1,2], Ladislava Řezníčková[1,2], Pavel Zahradníček[2,3], Petr Dobrovolný[1,2]

[1]Institute of Geography, Masaryk University, Brno, Czech Republic
5  [2]Global Change Research Institute, Czech Academy of Sciences, Brno, Czech Republic
[3]Czech Hydrometeorological Institute, Brno, Czech Republic

*Correspondence to*: Rudolf Brázdil (brazdil@sci.muni.cz)

**Abstract.** This paper presents an analysis of fatalities attributable to weather conditions in the Czech Republic during the 2000–2019 period. The database of fatalities deployed contains information extracted from *Právo*, a leading daily 10 newspaper, and *Novinky.cz*, its internet equivalent, supplemented by a number of other documentary sources. The analysis is performed for floods, windstorms, convective storms, rain, snow, glaze ice, frost, heat, and fog. For each of them, the associated fatalities are investigated in terms of annual frequencies, trends, annual variation, spatial distribution, cause, type, place, and time, as well as the sex, age, and behaviour of casualties. There were 1164 weather-related fatalities during the 2000–2019 study period, exhibiting a statistically significant falling trend. Those attributable to frost (31 %) predominated, 15 followed by glaze ice, rain and snow. Fatalities were at their maximum in January and December and at their minimum in April and September. Fatalities arising out of vehicle accidents (48 %) predominated in terms of structure, followed by freezing or hypothermia (30 %). Most deaths occurred during the night. Adults (65 %) and males (72 %) accounted for the majority of fatalities, while indirect fatalities were more frequent than direct ones (55 % to 45 %). Hazardous behaviour accounted for 76 %. According to the database of the Czech Statistical Office, deaths caused by exposure to excessive 20 natural cold are markedly predominant among five selected groups of weather-related fatalities and their numbers exhibit a statistically significant rise during 2000–2019. Police yearbooks of the fatalities arising out of vehicle accidents indicate significantly decreasing trends in the frequency of inclement weather patterns associated with fatal accidents, as well as a decrease in their percentage in annual numbers of fatalities. The discussion of results includes the problems of data uncertainty, comparison of different data sources, and the broader context.

## 25  1 Introduction

Natural disasters are accompanied not only by extensive material damage, but also by great loss of human life, facts easily derived from data held by re-insurance agencies (e.g. Munich RE, 2018; Swiss Re, 2019; Willis Re, 2019). It is therefore hardly surprising that this situation was also reflected in The Sendai Framework for Disaster Risk Reduction 2015–2030 (SFDRR), adopted at the Third UN World Conference on Disaster Risk Reduction in Sendai (Japan) on 18 March 2015. The 30  Framework suggested targets and priorities intended "to prevent new and reduce existing disaster risks" (UNODRR, 2020;




Wright et al., 2020). Events associated with climate and weather constitute highly important elements among natural disasters in terms of the damage they do and associated fatalities, as documented by the great number of papers addressing fatalities at world-wide or continental scales. For example, Ferreira et al. (2013) investigated the impact of development on flood fatalities in 92 countries between 1985 and 2008. Holle (2016) presented a broader, world-wide overview of lightning

fatalities. Gasparrini et al. (2017) investigated projections of excess temperature-related mortality for a range of climate-change scenarios. Vicedo-Cabrera et al. (2018) evaluated changes in heat- and cold-related mortality under scenarios consistent with the Paris Agreement targets. Franzke and Torelló i Sentelles (2020) analysed trends in weather and climate hazards based on continentally-aggregated fatality data in relation to large-scale climate variability.

Although Europe suffered only 8.9 % of world-wide climate- and weather-related fatalities in 1980–2017, far fewer than

Asia (71.1 %) and slightly fewer than North America (13.7 %) (Munich RE, 2018), Europe also has a very serious problem; particular attention has been paid to deaths associated with heat-waves and floods on this subcontinent. The exceedingly hot summer of 2003 (Fink et al., 2004; García-Herrera et al., 2010), when heat waves accounted for over 70,000 fatalities (especially in France, Germany, Italy, Portugal, Romania, Spain, and the UK; e.g. Robine et al., 2008), brought the problem into sharp focus. Heat waves and drought in the European part of Russia during July–September 2010 brought about *c.*

56,000 fatalities (Shmakin et al., 2013; Munich RE, 2014). Heat-wave-related mortality has been extensively investigated at national level for several European countries, among them, Slovakia (Vyberčki et al., 2015), Finland (Kim et al., 2018), Poland (Graczyk et al., 2019), and in western Europe overall (Vautard et al., 2020). In contrast, slightly less attention has been devoted to mortality during cold spells (e.g. Analitis et al., 2008; Ebi, 2015; Kinney et al., 2015; Psistaki et al., 2020). The impacts of climate change on future heat- and cold-related mortality in the Netherlands were evaluated by Botzen et al.

50 (2020).

Studies of flood-related fatalities, sometimes linked with those associated with landslides, tend to focus on southern Europe and the Mediterranean (e.g. Zêzere et al., 2014; Diakakis, 2016; Pereira et al., 2016, 2017; Diakakis and Deligiannakis, 2017; Salvati et al., 2018; Vinet et al., 2019; Grimalt-Gelabert et al., 2020), although other countries have their shares (e.g. Hilker et al., 2009 for Switzerland and Špitalar et al., 2020 for Slovenia). Petrucci et al. (2019b) presented the MEditerranean

Flood Fatalities (MEFF) database with detailed data concerning flood fatalities in five Mediterranean regions for the 1980–2015 period, later extended to nine regions for the 1980–2018 period as the EUropean Flood Fatalities (EUFF) database (Petrucci et al., 2019a).

Other weather hazards attract less attention. For example, lightning fatalities and injuries were studied for the United Kingdom by Elsom (2001) and Elsom and Webb (2014), and for Romania by Antonescu and Cărbunaru (2018). Haque et al.

(2016) investigated fatal landslides in Europe. Fatalities associated with a number of natural hazards in Switzerland were investigated by Badoux et al. (2016), while Heiser et al. (2019) presented a torrential event catalogue for Austria, but without particular attention to associated fatalities. Salvador et al. (2020) analysed the short-term effects of droughts on daily mortality in Spain. Some contributions confined themselves to only the most catastrophic events (e.g. Trigo et al., 2016; Diakakis et al., 2020).

In the Czech Republic, just as on the international scale, studies of fatalities associated with heat waves are the most frequent (e.g. Kyselý and Kříž, 2008; Kyselý and Plavcová, 2012; Knobová et al., 2014; Hanzlíková et al., 2015; Urban et al., 2017; Arsenović et al., 2019). However, attention has also been devoted to the fatal effects of cold spells. For example, Kyselý et al. (2009) and Plavcová and Urban (2020) analysed the impacts of compound winter extremes upon mortality rates. Brázdová (2012) worked on selected floods in the Czech Republic in order to develop a simple model for estimation of

flood-fatalities. Czech flood fatality data also appeared in the EUFF database and were worked upon by Petrucci et al. (2019a). Brázdil et al. (2019b) analysed the potential of documentary data in the study of weather-related fatalities and presented preliminary results for the 1981–2018 period.

    The first two decades of the 21st century make up the period that experienced the most profound temperature increases world-wide since records began, inclusive of the Czech Republic (e.g. Zahradníček et al., 2020). The general increase in

climatic and weather hazards raises the question as to whether this situation has also been reflected in the number of fatalities associated with weather phenomena. The current paper consists of an investigation and analysis of variability and existing trends in weather-related fatalities over the territory of the Czech Republic in the 2000–2019 period with respect to a selection of influencing factors. The work is based on its own mortality database created from newspapers and in comparison with other official/administrative sources of information. Section 2 describes the basic data sources used for analysis, while

the methodology appears in Section 3. The results in Section 4 describe weather-related fatalities, considered as a whole and for individual weather phenomena, as well as addressing those selected from official databases. The results are further discussed with respect to data uncertainty, comparison of different fatality information sources and the broader (central) European context in Section 5. A summary of the most important results appears in Section 6.

## 2 Data

### 2.1 Documentary data

    Newspapers and their websites are the most important source of documentary information for more recent times. They report not only upon political, socio-economic and social matters, but they also reflect considerable public interest in disasters, weather phenomena and associated damage and fatalities. This study gathers information from the print edition of the daily newspaper *Právo* and *Novinky.cz*, its internet equivalent. As well as its usual, wider coverage of national and international

news, it also appears in several different editions that dedicate space to particular regions of the Czech Republic, thus providing a highly useful source of fatality information. In addition to systematic reading of the newspaper, the research team employed the internet, entering a set of 52 key words (e.g. casualty, died, killed, black ice, flood, windstorm, lightning, frost, heat-wave) and 34 set phrases (e.g. wet road, slippery road, frost casualty, cold casualty, bad weather, bad visibility) to monitor *Právo* and *Novinky.cz* for further fatality events.

Individual newspaper reports differ in their style and approach to detail in descriptions of events resulting in fatalities. Although some cases are made immediately obvious within the headlines (e.g. "Lightning kills woman", "Icy road leads to





fatal car accident", "Seven homeless die of hypothermia, one frozen on park bench"), other casualties appear in the run of text, or remain quite general. Three examples of fatality records appear below:

(i) One of fatalities during disastrous August 2002 flood was a man in Putim (*Právo*, 15 August 2002, p. 2) (for localities in the Czech Republic see Fig. A1): "Recent disastrous floods have claimed their tenth human life. The latest was an 81-year-old man, drowned near Putim in the Písek region. […] The man, a permanent resident of Písek, was spending the summer in a cottage near Putim, which was threatened by huge quantities of water last Monday. Although rescue workers asked him repeatedly to leave the premises and evacuate, he refused. His corpse was found in the cottage yesterday [14 August]. He appeared to have fallen from the loft into the flooded [lower] house […]".

(ii) A woman died on 25 June 2008 during a thunderstorm in Svitavy (*Právo*, 27 June 2008, p. 1, 5): "According to the police, a woman was hurrying home with a group of friends on the evening of Wednesday [25 June]. As the first drops of rain started to fall, their group of six took shelter under the small roof of a garden restaurant in a town stadium. At that moment, the wind gusted upon an avenue of poplar trees with such force that 10-m branches were torn from it. One such branch struck a 47-year-old woman standing only 10 m from safety. It hit her neck. […] The emergency services took her [to hospital] within ten minutes but she succumbed to injuries of the spine."

(iii) A woman died in an accident on a snow-covered road in the Bruntál region on 20 April 2017 (Novinky.cz, 2020): "April snowfall in the Bruntál region has claimed its first road victim. Two private cars crashed. A 58-year-old female passenger died and a further three people were injured. The road was covered in snow at the time of the accident. According to the police, a Mitsubishi car got into a series of skids on the snowy road and crossed into the opposite lane, where it collided with a Škoda Fabia. The female victim was a passenger in the Fabia. She was taken to Ostrava hospital, but died shortly thereafter."

The basic set of fatality data from *Právo* and *Novinky.cz* was further verified, and sometimes supplemented, by reports from further documentary sources, such as other national or local newspapers, professional reports/papers or specialist articles, either published in print or available online.

## 2.2 Data from the Czech Statistical Office

Mortality yearbooks for the Czech Republic, arranged according to cause of death, sex and age are published by the Czech Statistical Office (CSO). They contain detailed summary data, specified with respect to various additional facts concerning any given year for the entire Czech Republic as well as its administrative units (CSO, 2020a, 2020b). Using the Office codes for cause of death employed on death certificates, the study herein considered: W00 – fall on ice or snow; X30 – exposure to excessive natural heat; X31 – exposure to excessive natural cold; X32 – exposure to solar radiation; X33 – lightning casualty; X36 – avalanche, landslide or other earth movement casualty; X37 – natural catastrophic storm casualty; X38 – flood (inundation) casualty; X39 – exposure to other and/or non-specified natural forces. Detailed CSO information about each death was gathered, including age, sex, level of education, place of permanent residence, date and code of death, supplemented from 2010 onwards by place of death.



**2.3 Czech police data**

The traffic police of the Czech Republic publish annual yearbooks of the accident rate on surface communications in the Czech Republic (PCR, 2020) which, among other matters, contain overviews of accidents in relation to weather conditions. This includes the numbers of fatalities occurring in normal weather conditions, during fog, the onset of rain and light rain, rain, snowfall, rime and glaze ice, gusty winds and other inclement weather conditions. A further table relates to fatalities with respect to the state of the road, where weather-related data include categories wet (road); glaze ice and snow – dusted; glaze ice and snow – non-dusted; continuous snow – slushy; sudden change (rime, glaze ice). Accidents are also categorised by visibility, specifying whether this deteriorated during the day due to the weather conditions (rain, fog) or during the night due to ambient conditions (with and without road lighting).

**3 Methods**

Fatality data from the *Právo* news outlets were first critically evaluated with respect to the quality and comprehensiveness of the reports. Those describing the circumstances leading to deaths, including detailed fatality data, were considered credible. Certain reports appeared in both the print version of *Právo* and in *Novinky.cz*, its internet equivalent. The latter tended to be more prompt in its reporting. Sometimes reports from the two sources complemented one another (e.g. by age of the deceased). Only fatal events that occurred within the territory of the Czech Republic were considered; weather-related deaths of Czech residents happening abroad were not taken into account.

The data extracted were used to create a database of Czech weather-related fatalities, applying the structure of the MEFF database by Petrucci et al. (2019b). For each fatality, this includes (i) date; (ii) locality; (iii) type of weather event (see below); (iv) part (hour) of the day (morning 0400–0800 CET, forenoon 0800–1200 CET, afternoon 1200–1800 CET, evening 1800–2200 CET, night 2200–0400 CET); (v) name of the casualty; (vi) sex (male, female); (vii) age (exact in years or estimated: child 0–15 years, adult 16–65 years, elderly 66 years or more); (viii) cause of death (drowning, falling tree/branch, vehicle accident, underlying health reason, freezing to death/hypothermia, lightning strike, other reason); (ix) place of death (river/lake/reservoir/bank, within a building, road, open space in a built-up area, countryside, other place); (x) type of fatality (direct, indirect); (xi) behaviour (non-hazardous and hazardous); and (xii) source of information. If any of these items was not available, the corresponding entry was categorised as "unknown (absent)".

An example of the interpretation of a fatality report reads: "On 4 July 2009 [date] at *c*. 1800 hours [time] at Benešov nad Ploučnicí [locality], 40-year-old [age] woman [sex] Naděžda Rubnerová [name] drowned [cause of death] in the River Ploučnice [place of death] during a flash flood [event]. She was carried away by a torrent of water [direct casualty] while helping to rescue a wheelchair-bound woman [non-hazardous behaviour]." (Idnes.cz, 2020) [source].

The term "weather-related fatalities" was associated with the events below:

(i) Flood: This includes floods arising out of single-day or multi-day rainfall during precipitation-rich synoptic situations (rainy floods), of sudden melting of deep snow cover (snow floods) and of a combination of snow-melt and rainfall,





sometimes with ice jams on the rivers (mixed floods) on the one hand, and flash floods arising from cloudbursts or torrential rains during thunderstorms on the other.

(ii) Windstorm: Strong winds resulting from large horizontal gradients of air pressure, lasting from a few hours to some
165 days, are considered windstorms.

(iii) Convective storm: This includes phenomena associated with the development of cumulonimbus cloud, such as very strong wind (e.g. squall, tornado, downburst), lightning strike, downpour, and hail.

(iv) Rain: This includes, in particular, rain and wet street communications surfaces/tracks.

(v) Snow: This includes, in particular, cases of snow calamity and avalanche.

170 (vi) Glaze ice: This includes ice-patches or glaze-ice cover on streets, roads and communications.

(vii) Frost: This includes severe frosts occurring as a part of cold spells as a cause of death, as well as fatalities involving bodies of water insufficiently frozen for the activity undertaken on ice.

(viii) Heat: This includes extremes of high temperature occurring in the course of heat waves.

(ix) Fog: This includes cases of significantly decreased horizontal visibility due to fog.

175 With respect to the limited length of the series of weather-related fatalities (20 years) compiled, and due to deviations from normality in data distribution within certain categories, trend analysis of fatalities was based on two approaches. The first employed a simple regression model based on least-squares estimate and evaluation of the statistical significance of slopes, based on the t-test. The second – nonparametric – approach evaluated the significance of the trend by means of the Mann-Kendall (Kendall, 1975) test and Sen's method for assessment of the magnitude of the trend. In the latter approach, the data
180 need not conform to any particular distribution. These two different methods were considered sufficient to provide a robust estimate of trends.

Fatalities in each group (i)–(ix) were analysed in detail in terms of their annual numbers, with linear trend estimated from least squares (statistical significance was set at the level of 0.05), of annual variations, spatial distribution and structure according to cause of death, type of death, place of death, time of death, age, sex and behaviour (Figs. 1–9). The same was
185 performed for all groups together (Fig. 10) and separately for fatalities in vehicle accidents (Fig. 11). Some of these characteristics were considered in analyses based on the official demographic databases of the CSO (Fig. 12) and/or the police database of vehicle accidents (Fig. 13).

Numbers of fatalities were presented at monthly, seasonal (winter – DJF, spring – MAM, summer – JJA, autumn – SON), half-year (winter – October to March, summer – April to September), and annual levels.



# 4 Results

## 4.1 Fatalities in individual weather events

### 4.4.1 Flood

During the 2000–2019 period, floods in the Czech Republic contributed to the deaths of 112 people, an average of 5.6 fatalities a year, of which 39 (34.8 %) were attributed to flash floods. The maximum of 21 fatalities occurred in 2002, associated with an exceedingly heavy rainy flood in August in Bohemia (17 fatalities), followed by 18 fatalities in 2010 and 2013, when further extraordinary floods struck the Czech Republic (Fig. 1a). No flood fatalities occurred in 2003, 2004 and 2016. The numbers of fatalities exhibit a statistically insignificant falling trend. In terms of annual distribution, the maximum number of fatalities appeared in June (33.9 %), followed by August (25.0 %), with a high proportion of flash-flood-related fatalities during JJA (Fig. 1b). Spatial distribution indicated that flood-related fatalities were more frequent in Bohemia, the western part of the Czech Republic, than in Moravia, its eastern part, where a higher concentration of fatalities was recorded in eastern Moravia (Fig. 1c). While 83.0 % of casualties drowned, 10.7 % died due to health problems (e.g. heart failure during rescue work) (Fig. 1d). A total of 82.1 % of fatalities were evaluated as "direct" (Fig. 1e). As might well be expected, 70.5 % of the fatalities took place in running water or close it; 16.1 % died in collapsing buildings (Fig. 1f). Despite the fact that the times of death were not specified for over half the fatalities, a local maximum appeared in the evening (Fig. 1g). In the demographic structure of fatalities, adults (58.0 %) and males (73.2 %) clearly predominated (Fig. 1hi). Non-hazardous behaviour among flood fatalities was more prevalent than hazardous (43.8 % to 38.4 %) (Fig. 1j).

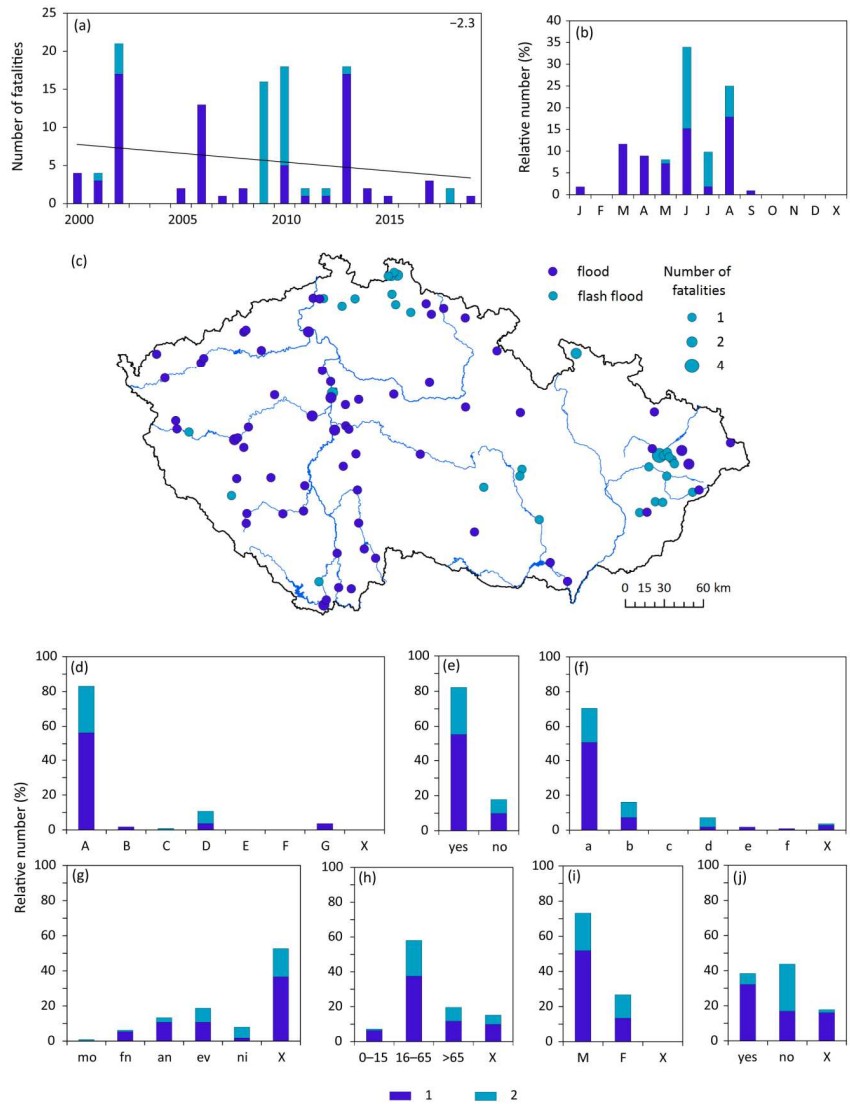

**Figure 1. Characteristics of flood fatalities (1 – flood, 2 – flash flood) in the Czech Republic during the 2000–2019 period: a) fluctuation with linear trend (top right, in fatalities/10 years); b) annual variation; c) spatial distribution (two fatalities lack exact locations); d) cause of death; e) type of death; f) place of death; g) part of the day; h) age; i) sex; j) behaviour. Symbols: A – drowning, B – tree/branch fall, C – vehicle accident, D – underlying health reason, E – freezing to death/hypothermia, F – lightning strike, G – other reason; a – river/lake/reservoir/bank, b – within a building, c – road, d – open space in built-up area, e –**



**open countryside, f – other place; mo – morning, fn – forenoon, an – afternoon, ev – evening, ni – night; M – males, F – females; X – unknown.**

**4.1.2 Windstorm**

Windstorms claimed 37 fatalities in the Czech Republic during the 2000–2019 period, an average of 1.8 fatalities a year. While nine fatalities were recorded in 2017 (four people died as a result of Storm Herwart on 29 October), no windstorm-related fatalities were recorded for nine years (in particular between 2012 and 2019, apart from 2017) (Fig. 2a). The infamous Storm Kyrill was responsible for six deaths on 18–19 January 2007. Windstorm-related fatalities exhibit a

statistically insignificant falling linear trend. In terms of annual variation, the highest proportion of fatalities appeared in October (21.6 %), followed by January (18.9 %); 81.1 % of all windstorm-related fatalities occurred in the winter half-year (Fig. 2b). The geographical distribution of fatalities indicated higher occurrence in the southern part of the Czech Republic, in central and northern Bohemia (Fig. 2c). More than half the fatalities (56.8 %) resulted from falling trees or their branches, and nearly a fifth (18.9 %) were attributable to other reasons, such as a falling roof or a person falling from a damaged roof

during attempted repairs; 10.8 % drowned in strong-wind-related boating incidents on lakes and reservoirs (Fig. 2d). A total of 86.5 % of fatalities were interpreted as "direct" (Fig. 2e). Among places of death, open spaces in built-up areas (35.1 %), roads (24.3 %) and the open countryside (21.6 %) were reported (Fig. 2f). Most fatalities occurred in the afternoon (37.8 %), but time of day was absent for 32.4 % (Fig. 2g). Adults (75.7 %) and males (83.8 %) clearly predominated among reported fatalities (Fig. 2hi). The percentage of fatalities involving non-hazardous behaviour was significantly higher than that for

hazardous (67.6 % to 27.0 %) (Fig. 2j).


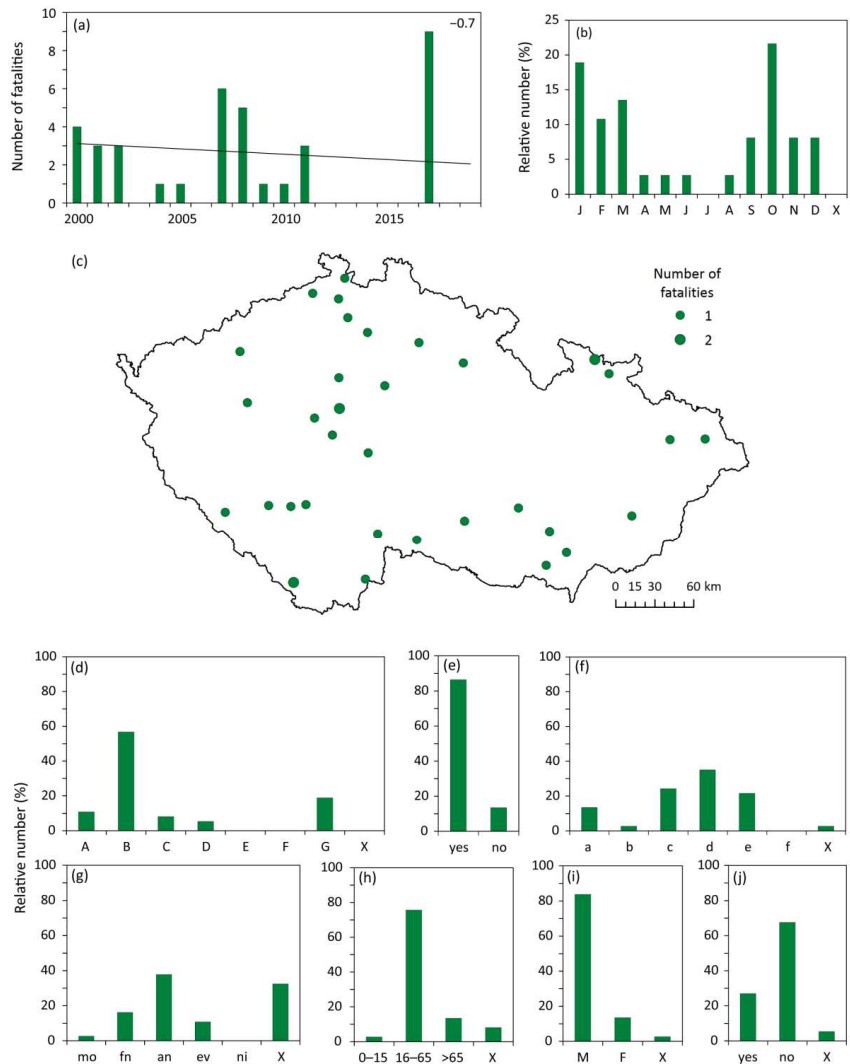

**Figure 2. Characteristics of windstorm-related fatalities in the Czech Republic during the 2000–2019 period: a) fluctuation with linear trend (top right, fatalities/10 years); b) annual variation; c) spatial distribution; d) cause of death; e) type of death; f) place of death; g) part of the day; h) age; i) sex; j) behaviour. For explanation of symbols see Fig. 1.**



### 4.1.3 Convective storm

A total of 46 fatalities were attributed to convective storms in the Czech Republic during the 2000–2019 period, an average of 2.3 fatalities a year: 17 (37.0 %) were caused by strong winds, 15 (32.6 %) by lightning strikes, eight (17.4 %) by rain or hail and six (13.0 %) were specified as only "during a thunderstorm". Fatalities reached a maximum of six in 2007 and 2008, while none were recorded in 2011–2013 and 2016 (Fig. 3a). The associated falling linear trend of 1.7 fatalities/10 years was statistically significant at the 0.05 level. All the fatalities recorded occurred during the summer half-year, with a maximum in June (32.6 %), followed by July (21.7 %) (Fig. 3b). The geographical distribution features a higher concentration over Bohemia in a belt extending from the south-west to the north-east (Fig. 3c). Falling trees or their branches and lightning strikes led to equal numbers of deaths (30.4 % each), followed by vehicle accidents at 19.6 % (Fig. 3d). A total of 63.0 % of all these fatalities were classified as "direct" (Fig. 3e). Among places of death, roads (30.4 %), open spaces in built-up areas (23.9 %) and open countryside (19.6 %) were the most frequent (Fig. 3f). Most fatalities occurred in the afternoon (41.3 %) (Fig. 3g). Adults (78.3 %), males (60.9 %) and hazardous behaviour (52.2 %) predominated in fatalities resulting from convective storms (Fig. 3hij).

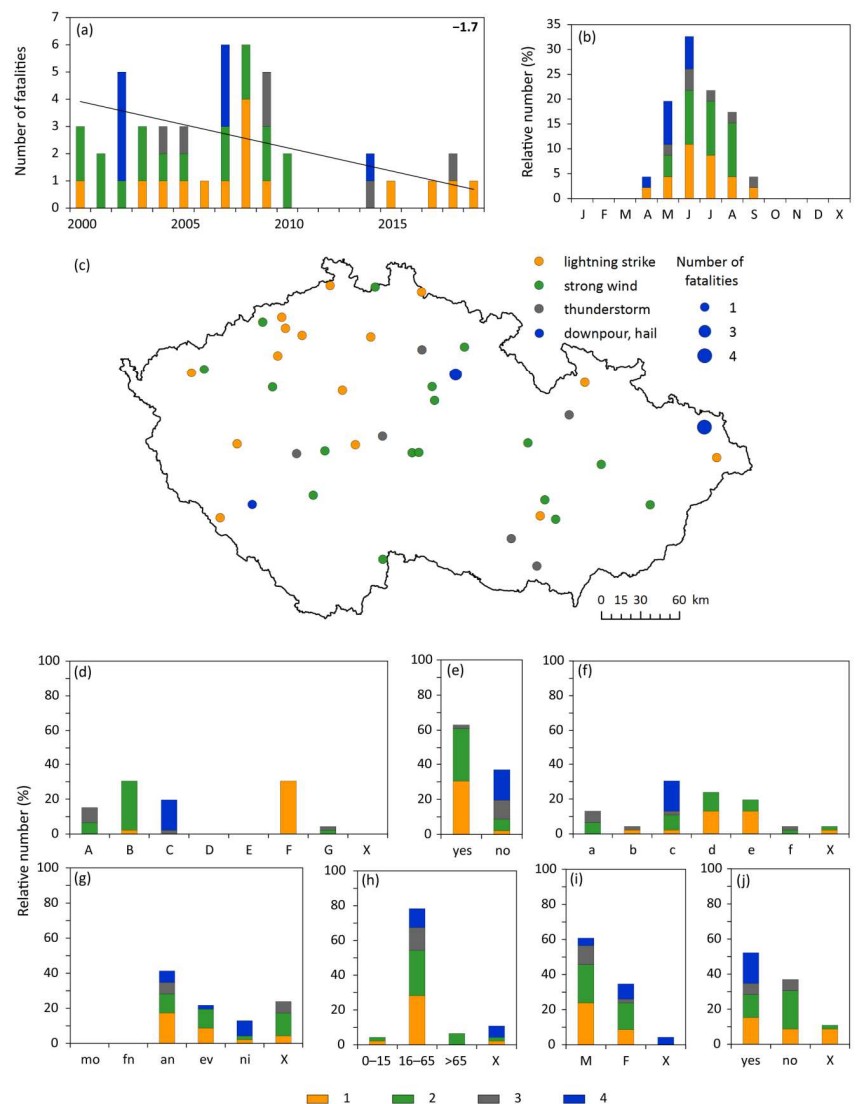

**Figure 3. Characteristics of convective-storm-related fatalities (1 – lightning strike, 2 – strong wind, 3 – thunderstorm, 4 – downpour, hail) in the Czech Republic during the 2000–2019 period: a) fluctuation with linear trend (top right, fatalities/10 years); b) annual variation; c) spatial distribution; d) cause of death; e) type of death; f) place of death; g) part of the day; h) age; i) sex; j) behaviour. For explanation of symbols see Fig. 1.**






### 4.1.4 Rain

Rain led to 205 fatalities in the Czech Republic during the 2000–2019 period, an average of 10.2 fatalities a year. A
maximum of 26 fatalities occurred in 2001, followed by 23 in 2014; only two occurred in 2003 and 2004 (Fig. 4a). A falling
but statistically insignificant trend was evident. In terms of annual distribution, nearly half the fatalities (49.8 %) occurred in
the summer months, with the highest proportion in July (21.5 %). Proportions in the months of the other three seasons did
not rise above 10 % (Fig. 4b). Rain-related fatalities were distributed over the whole Czech Republic, with a higher
concentration in some of the smaller regions and lower frequency near borders, for example, north-western, south-western
and southern Bohemia and south-western Moravia (Fig. 4c). All these fatalities were classified as indirect consequences of
vehicle accidents. They tended to occur in the afternoon (26.3 %) (Fig. 4d). In terms of structure, adults made up 68.3 % of
fatalities (age not reported for 20 %) and males 65.9 % (Fig. 4ef). A total of 97.6 % the fatalities were classified as arising
out of hazardous behaviour on the part of the victims, or of the driver(s) responsible for the accident.

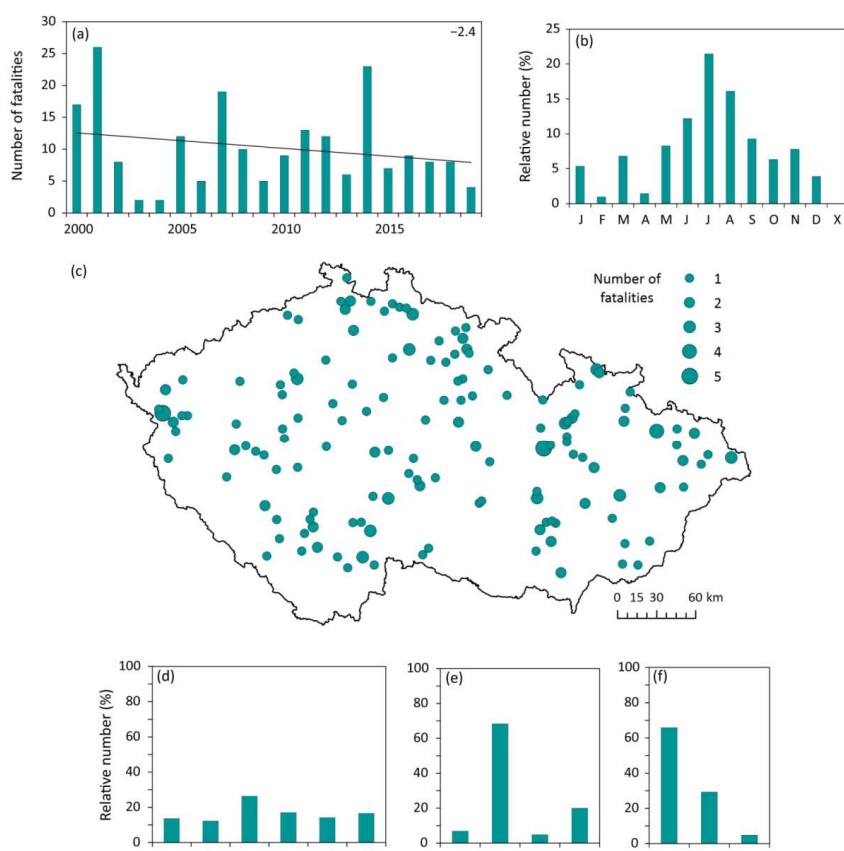

**Figure 4. Characteristics of rain-related fatalities in the Czech Republic during the 2000–2019 period: a) fluctuation with linear trend (top right, fatalities/10 years); b) annual variation; c) spatial distribution (four fatalities lack exact location); d) part of the day; e) age; f) sex. For explanation of symbols see Fig. 1.**

### 4.1.5 Snow

A total of 137 fatalities in the Czech Republic during the 2000–2019 period may be attributed to snow, an average of 6.8 fatalities a year. While a maximum of 16 snow-related fatalities occurred in 2005 (15 in 2006), none was recorded in 2014 (Fig. 5a). These data also include six fatalities (4.4 %) in avalanches (two in 2010, one each in 2006, 2008, 2009 and 2015). A falling linear trend reached 4.5 fatalities/10 years and was statistically significant. As is to be expected, snow-related fatalities occurred only in the months of the winter half-year (with the exception of a single fatality in April), with maxima in January (34.3 %) and February (29.2 %); the percentages for November and December were identical, at 12.4 % (Fig. 5b). In



terms of spatial distribution, snow-related fatalities tended to concentrate into a large number of smaller areas, while in some
larger regions no localities with casualties were recorded at all (Fig. 5c). Vehicle accidents were involved in a total of 84.7 %
of the fatalities (Fig. 5d), thus 94.9 % of them were classified as "indirect" (Fig. 5e). The percentage of people dying on the
roads achieved 81.0 % (Fig. 5f); more than 5 % occurred in the built-up areas and in open countryside (avalanche casualties).
While the time of day at which death took place remained unknown for 38.0 % of fatalities, the morning and afternoon

proportions were 18.2 % and 17.5 % respectively (Fig. 5g). Despite the relatively high percentage of fatalities for which age
was unspecified (29.2 %), 51.1 % were categorised as "adult" (Fig. 5h). Males accounted for 59.1 % of snow-related
fatalities, and hazardous behaviour for 88.3 % (Fig. 5ij).



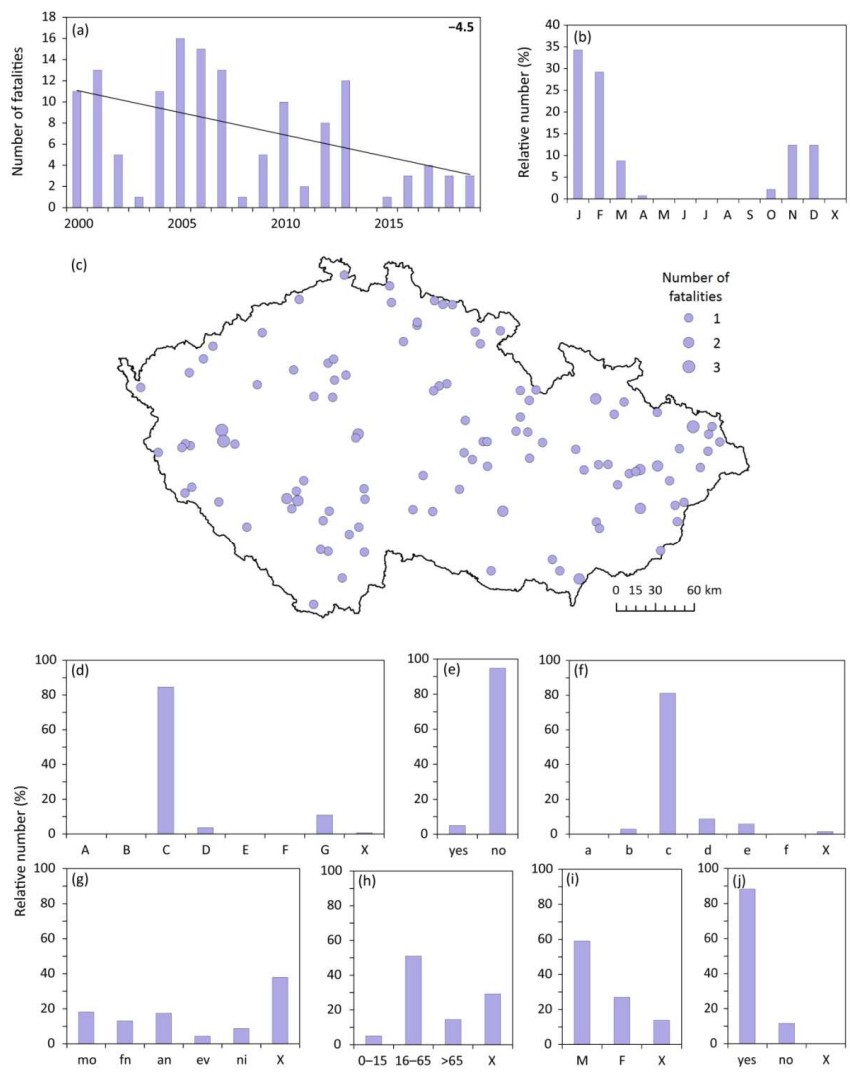

**Figure 5. Characteristics of snow-related fatalities in the Czech Republic during the 2000–2019 period: a) fluctuation with linear
trend (top right, fatalities/10 years); b) annual variation; c) spatial distribution (12 fatalities lack exact location); d) cause of death;
e) type of death; f) place of death; g) part of the day; h) age; i) sex; j) behaviour. For explanation of symbols see Fig. 1.**






### 4.1.6 Glaze ice

Glaze ice was responsible for 222 fatalities in the Czech Republic during the 2000–2019 period, an average of 11.1 fatalities a year. With a maximum of 24 fatalities in 2005 (23 in 2004) and a minimum of two fatalities in 2008, the 20-year series is
characterised by a statistically significant falling linear trend of 7.1 fatalities/10 years (Fig. 6a). In terms of annual variation, glaze-ice fatalities occurred between October and April with a maximum in December (33.8 %), followed by January with 27.0 % (Fig. 6b). They occurred over the whole territory with higher concentrations in some areas, lower or none in others (Fig. 6c). Vehicle accidents were involved in a total of 95.0 % of glaze-ice fatalities (Fig. 6d), i.e. indirect casualties. The place of death was a road for 87.8 % of fatalities, and 9.0 % took place on communications in built-up areas (Fig. 6e).
Although exact time of death remained unknown for nearly a fifth of the fatalities, 36.5 % occurred during the morning (Fig. 6f). Adults (64.0 %) and males (66.7 %) predominated in the structure of fatalities, together with hazardous behaviour (96.4 %) on the part of the victims or those responsible for fatal accidents (Fig. 6ghi).



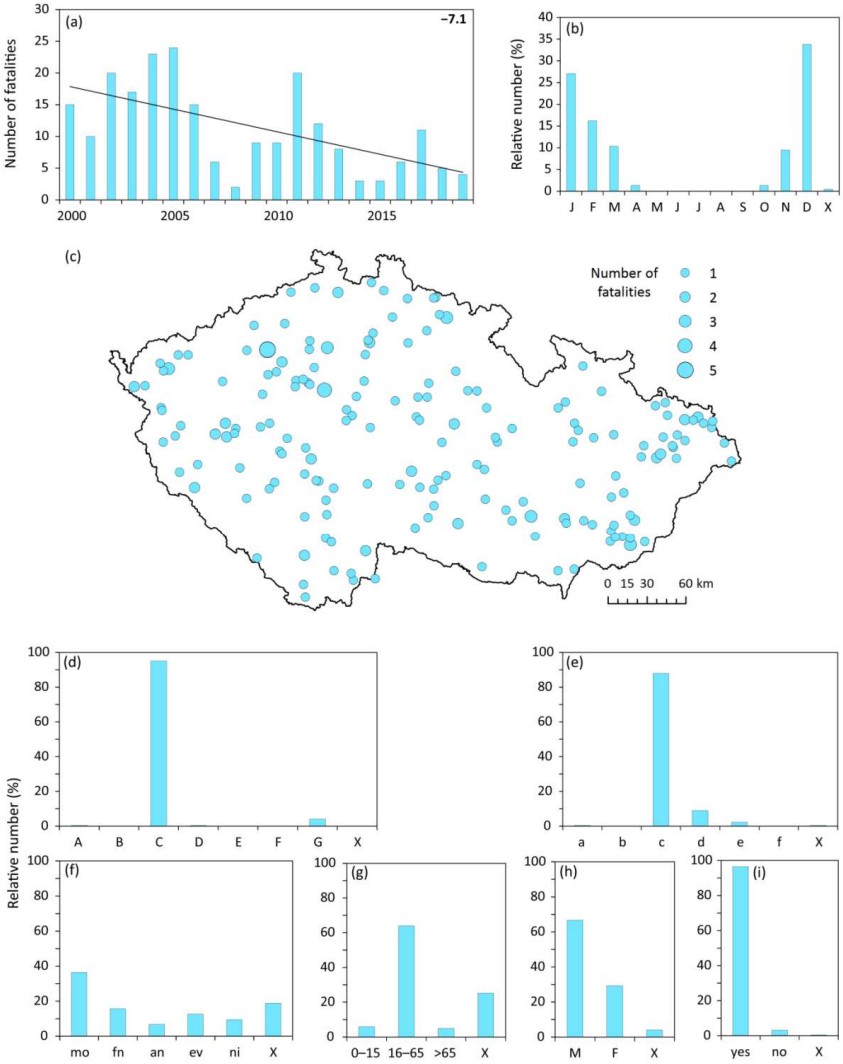

**Figure 6. Characteristics of glaze-ice-related fatalities in the Czech Republic during the 2000–2019 period: a) fluctuation with linear trend (top right, fatalities/10 years); b) annual variation; c) spatial distribution (eight fatalities lack exact location); d) cause of death; e) place of death; f) part of the day; g) age; h) sex; i) behaviour. For explanation of symbols see Fig. 1.**



### 4.1.7 Frost

A total of 360 frost-related fatalities rendered it the largest category of all those analysed in the Czech Republic during the 2000–2019 period, providing an average of 18 fatalities a year. Maximum numbers were disclosed in 2010 (52) and in 2012 (42), while in 2013 and 2015 only three were recorded (Fig. 7a). However, the relevant falling linear trend of 5.8 fatalities/10 years was statistically insignificant. The highest percentage of frost-related fatalities appeared in January (34.7 %) and December (28.6 %). Beyond the months of the winter half-year, only one fatality was recorded, in April (0.6 %) (Fig. 7b). Since the majority of deaths occurred among the homeless, fatalities were partly concentrated into large cities/towns such as Prague (79), Ostrava (24) and Brno (15) (Fig. 7c). While most succumbed to freezing or hypothermia (96.4 %), the remainder concerned cases in which the ice on ponds, reservoirs or rivers was insufficient for the activity undertaken and people fell through it and drowned (Fig. 7d). A similar proportion (96.1 %) of fatalities were characterised as "direct" (Fig. 7e). Open spaces in built-up areas accounted for 48.3 % in place of death, followed by open countryside (18.6 %) (Fig. 7f). Exact times of death were difficult to establish, remaining unknown for 65.5 % of fatalities, while for 29.2 % it was specified as occurring "at night" (Fig. 7g). Adults (66.9 %) and males (83.3 %) predominated among the further characteristics, followed by hazardous behaviour (68.1 %, but with 28.1 % unknown) (Fig. 7hij).

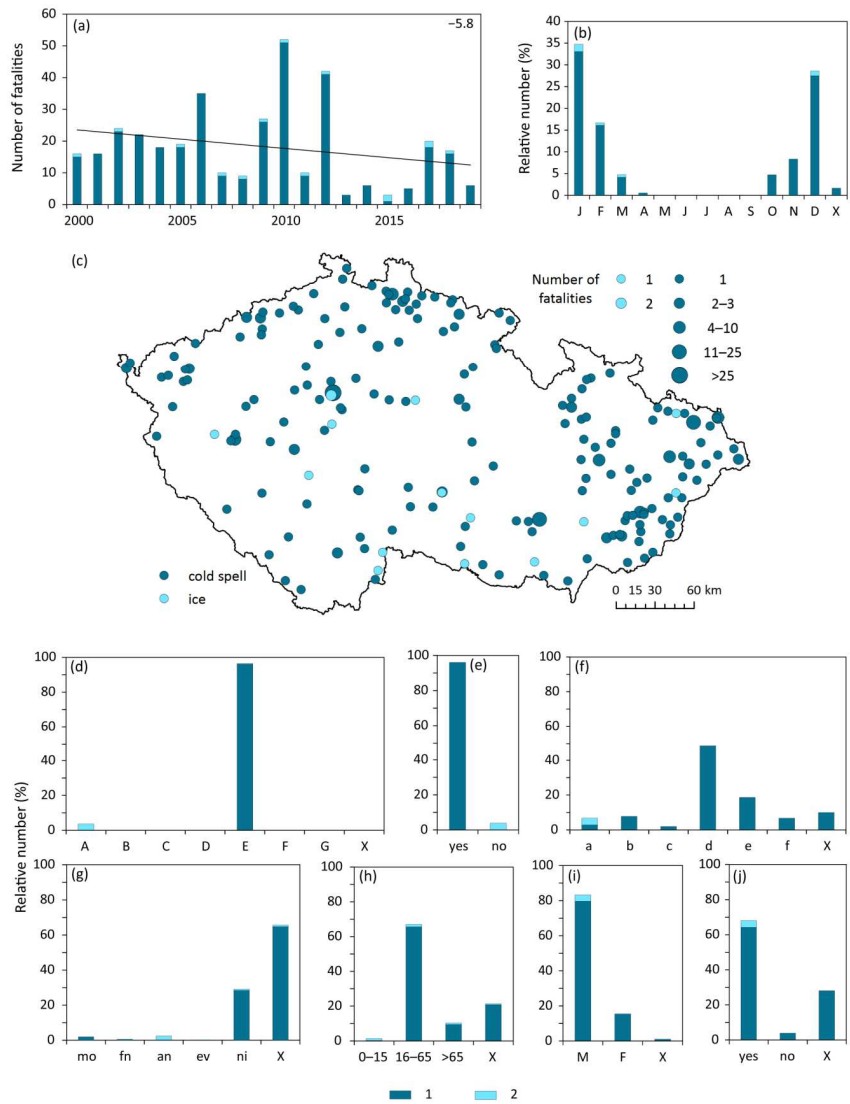

**Figure 7. Characteristics of frost-related fatalities (1 – cold spell, 2 – ice) in the Czech Republic during the 2000–2019 period: a) fluctuations with linear trend (top right, fatalities/10 years); b) annual variation; c) spatial distribution (six fatalities lack exact location); d) cause of death; e) type of death; f) place of death; g) part of the day; h) age; i) sex; j) behaviour. For explanation of symbols see Fig. 1.**





### 4.1.8 Heat

Only 20 fatalities in the Czech Republic were attributed to heat (heat waves) during the 2000–2019 period, an average of one fatality a year. Nine such cases were recorded in 2006, while in seven of the years only one fatality occurred, and none at all in 10 of the years. This is reflected in a statistically insignificant falling linear trend (Fig. 8a). Fatalities appeared only from May to August, with a maximum in June at 45.0 % (Fig. 8b). The spatial distribution of such a low number of fatalities reveals no features worthy of mention (Fig. 8c). Heart failure appears as the main cause of death, classified as "direct". A total of 40.0 % of fatalities occurred in built-up areas, but place of death was not specified in records for 35.0 % (Fig. 8d). The time of day was not entered for 70.0 % of the fatalities (Fig. 8e). Despite the fact that deaths in the "adult" category remained the highest (45.0 %), the percentage of the elderly was significantly high (25.0 %), the same figure as the "unknown" category (Fig. 8f). Males made up 70.0 % of fatalities according to sex (Fig. 8g) and the behaviour of 50.0 % of the fatalities was interpreted as "non-hazardous" (Fig. 8h).

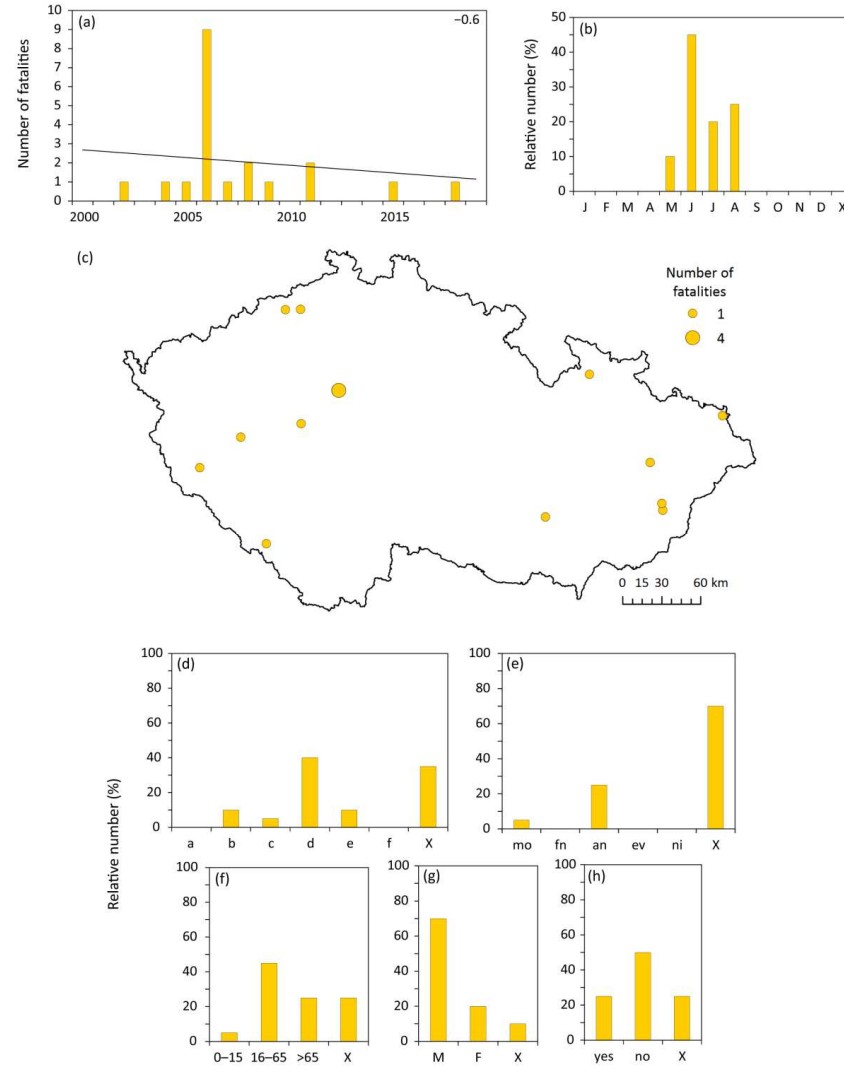

**Figure 8. Characteristics of heat-related fatalities in the Czech Republic during the 2000–2019 period: a) fluctuation with linear trend (top right, fatalities/10 years); b) annual variation; c) spatial distribution (four fatalities lack exact location); d) place of death; e) part of the day; f) age; g) sex; h) behaviour. For explanation of symbols see Fig. 1.**





### 4.1.9 Fog

Fog was responsible for 18 fatalities in the Czech Republic during the 2000–2019 period, an average of 0.9 fatalities a year. While five fatalities occurred in 2014, no such event was recorded for nine of the years (Fig. 9a). The rising linear trend was statistically insignificant. While 77.8 % of the fatalities occurred in SON, with a maximum in September (33.3 %), the

remainder were recorded in the winter months (Fig. 9b). The geographical distribution for such a small number of cases is of a somewhat random character (Fig. 9c). Decreased visibility was evident in the 88.9 % of fatalities attributable to vehicle accidents; two fatalities (11.1 %) occurred in an aeroplane crash (Fig. 9d). These indirect casualties took place on roads (83.3 %), also partly in the countryside (aeroplane crash) and in built-up areas (Fig. 9e). Half the fatalities occurred during the morning (Fig. 9f). The predominance of adults (83.3 %), males (66.7 %) and hazardous behaviour (94.4 %) characterised

other features of fog-related fatalities (Fig. 9ghi).


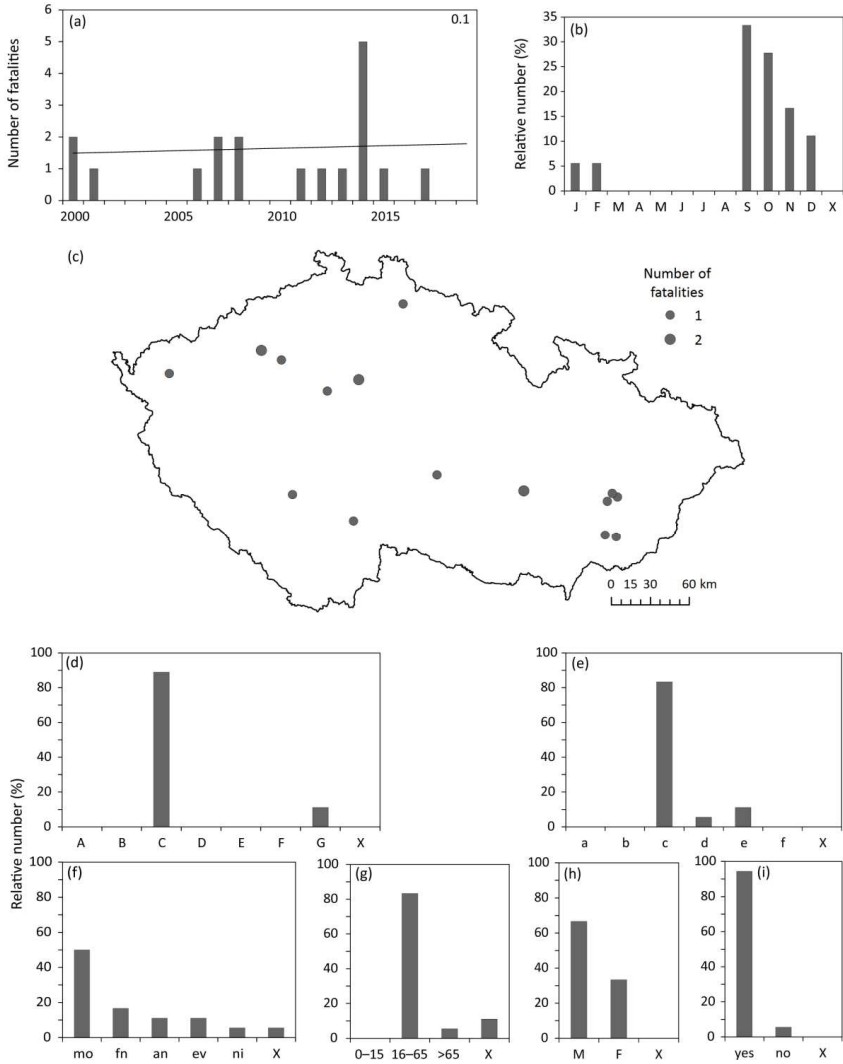

**Figure 9. Characteristics of fog-related fatalities in the Czech Republic during the 2000–2019 period: a) fluctuation with linear trend (top right, fatalities/10 years); b) annual variation; c) spatial distribution; d) cause of death; e) place of death; f) part of the day; g) age; h) sex; i) behaviour. For explanation of symbols see Fig. 1.**



### 4.2 Weather-related fatalities in total

A total of 1164 weather-related fatalities were recorded in the Czech Republic during the 2000–2019 period, an average of 58.2 fatalities a year. The maximum of 103 fatalities was recorded in 2010, the minimum of 18 in 2015 (19 in 2019) (Fig. 10a). A falling trend of 25.0 fatalities/10 years proved statistically significant at the significance level of 0.05. Almost a third of the fatalities (30.9 %) were taken up by frost-related cases, followed by glaze ice (19.1 %), rain (17.6 %) and snow (11.8 %). While floods were responsible for 9.6 % of fatalities, other weather factors stood at well below 5 %. These proportions influence the annual distribution of fatalities, bringing the maxima to January (21.7 %) and December (18.0 %), while minimum fatalities was recorded in April (1.9 %) and September (2.7 %) (Fig. 10b). A clear predominance of fatalities appears in DJF (52.5 %), while JJA shows a slightly higher proportion (20.0 %) compared with MAM and SON (12.5 % and 15.0 % respectively). A more marked difference appears in comparison of fatalities during the winter and summer half-years: 72.0 % compared with 28.0 % respectively. Distribution of all weather-related fatalities over the territory of the Czech Republic features its highest concentrations in several areas, such as southern, central and northern Bohemia and eastern Moravia and Silesia. Because of the high percentage of casualties arising out of vehicle accidents, certain major highways/roads are also clearly distinguishable. In terms of individual locations, Prague predominates with 95 fatalities, followed by Ostrava (31), Brno (23) and Plzeň (21) (Fig. 10c).

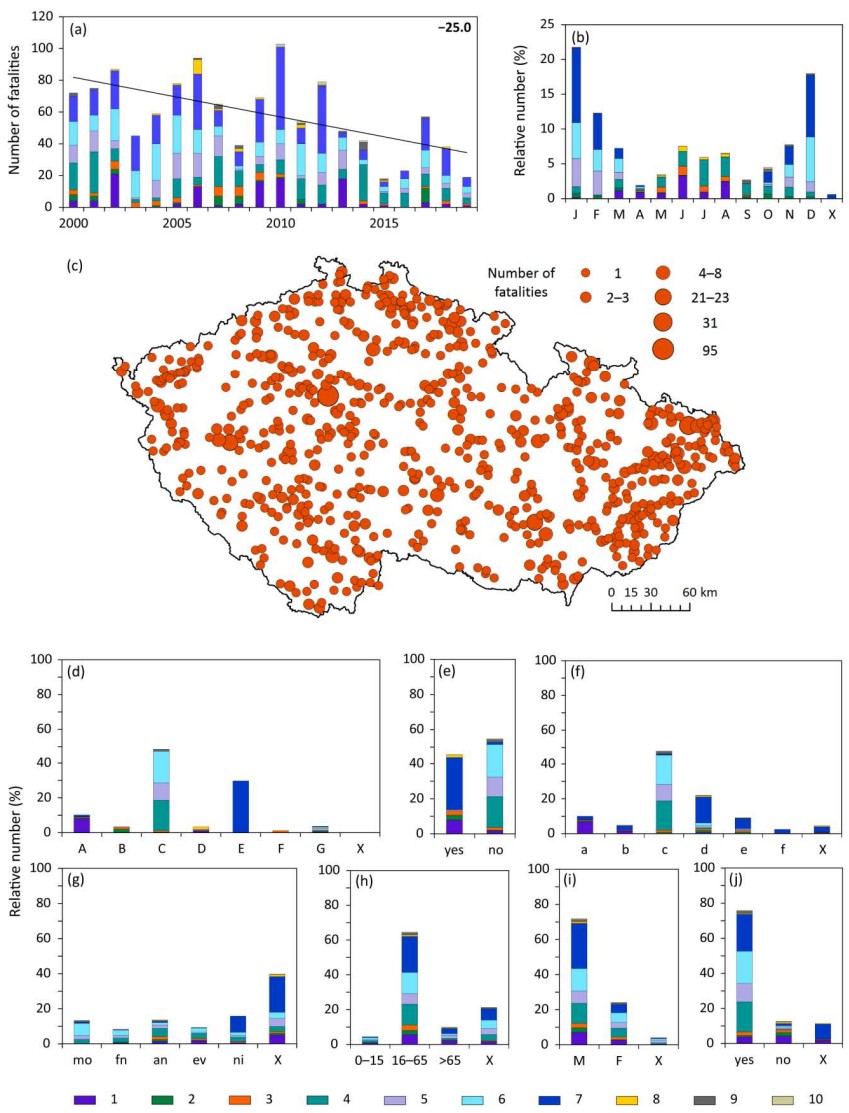


**Figure 10. Characteristics of weather-related fatalities (1 – flood, 2 – windstorm, 3 – convective storm, 4 – rain, 5 – snow, 6 – glaze ice, 7 – frost, 8 – heat, 9 – fog, 10 – other) in the Czech Republic during the 2000–2019 period: a) fluctuation with linear trend (top right, fatalities/10 years); b) annual variation; c) spatial distribution (36 fatalities lack exact locations); d) cause of death; e) type of death; f) place of death; g) part of the day; h) age; i) sex; j) behaviour. For explanation of symbols see Fig. 1.**





Vehicle accidents are the most frequent causes of death (48.4 %), followed by those associated with cold spells leading to freezing or hypothermia (29.8 %); drowning takes up 10.1 % of fatalities (Fig. 10d). Weather-related indirect fatalities are more frequent than those classified as direct (54.8 % to 45.2 % respectively) (Fig. 10e). The majority of fatalities occur on roads and communications (47.4 %), followed by open spaces in built-up areas (21.9 %). The areas around running water and reservoirs or in open countryside take their toll (10.0 % and 9.1 % respectively) (Fig. 10f). Available data indicates that

most deaths occur by night, but 39.7 % of fatalities were not attributed to any particular part of the day (Fig. 10g). In the structure of fatalities, the numbers of adults (64.6 %) and males (71.9 %) clearly predominate (Fig. 10hi). Over three-quarters of fatalities (75.9 %) may be attributed to hazardous behaviour on the part of actual casualties or that of other people immediately responsible for their deaths (Fig. 10j).

Of the total of 1164 weather-related fatalities, 66 (5.7 %) were identified as non-Czechs, in the country on business, holiday

or only in transit. Among these people, those from neighbouring Slovakia (17 fatalities), Poland and Germany (10 fatalities each) suffered most (56.1 % altogether). In addition to 12 other nationalities, 13 non-Czechs lacked exact specification of nationality. Nearly a third of them (30.3 %) fell victim to frost (e.g. the homeless), while the greatest part (51.5 %) died in vehicle accidents taking place during the occurrence of glaze ice, rain and snow.

Because vehicle accidents make up nearly half the weather-related fatalities in the Czech Republic during the 2000–2019

period, they were subjected to further, separate analysis. A total of 563 such fatalities (an average of 28.2 fatalities a year) were distributed among five types of weather event: glaze ice – 211 fatalities (37.5 %), rain and wet roads – 204 (36.2 %), snow – 116 (20.6 %), fog – 16 (2.8 %), and other events – 16 (2.8 %), of which nine fatalities (1.6 %) were generally associated with thunderstorms. The maximum of 51 fatalities was recorded in 2005, the minimum of 10 in 2015. The data exhibits a statistically significant falling trend of 13.4 fatalities/10 years (Fig. 11a). Among individual events, falling linear

trends were statistically significant for glaze ice and snow. Annual variation exhibits two maxima, primarily in the winter months, arising out of glaze ice and snow (January 19.2 %) and secondarily in the summer months in response to rain (July 7.8 %) (Fig. 11b). In the spatial distribution of fatalities, concentrations around main roads/highways, or certain parts of them, together with the main routes out of the country, are apparent (Fig. 11c). Using the vehicle accident casualties classified within "indirect deaths" and "hazardous behaviour" (96.8 %), 94 % of them died on roads and the remaining 6 %

in built-up areas and the countryside. Although the times day at which death occurred are absent for 21.1 % of fatalities, they were for the greater part recorded in the morning (24.7 %) (Fig. 11d). Adults made up 63.9 % of fatalities and numbers of deaths among the elderly were slightly higher than those for children (6.3 % and 5.3 % respectively) (Fig. 11e). There were more than twice as many male fatalities as female (64 % and 29 % respectively) (Fig. 11f).


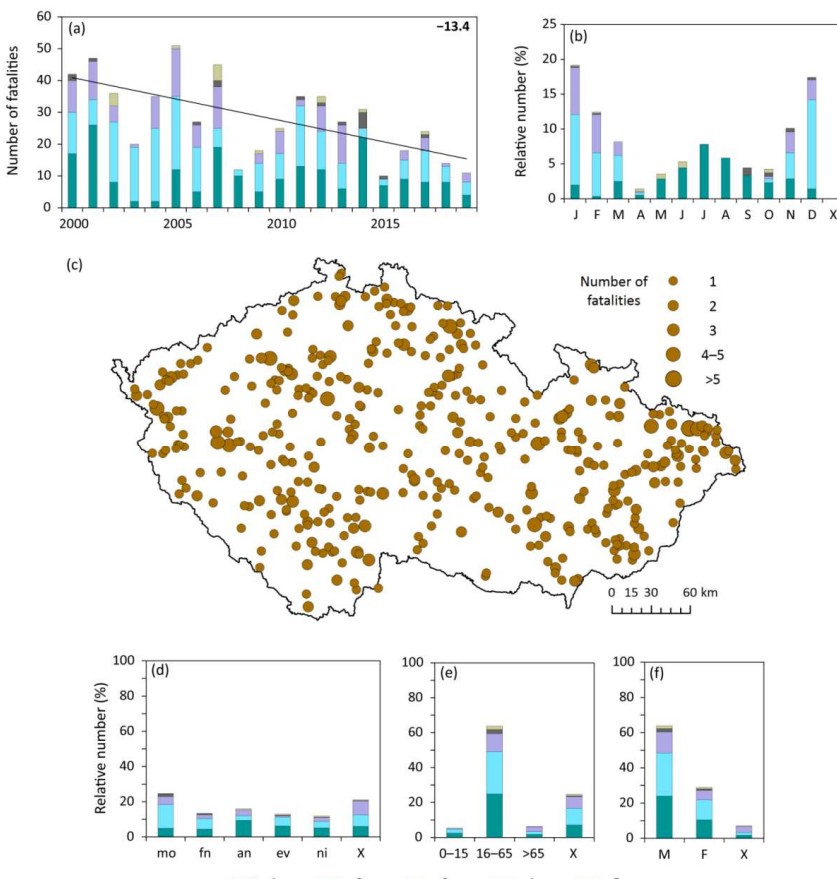

**Figure 11. Characteristics of vehicle accident fatalities with regard to weather events (1 – rain and wet roads, 2 – glaze ice, 3 – snow, 4 – fog, 5 – other inclement weather) in the Czech Republic during the 2000–2019 period: a) fluctuation with linear trend (top right, fatalities/10 years); b) annual variation; c) spatial distribution (21 fatalities lack exact locations); d) part of the day; e) age; f) sex. For explanation of symbols see Fig. 1.**

**4.3 Weather-related fatalities according to official data**

**4.3.1 Demographic data**

Fluctuations in fatalities associated with selected causes of death (as specified and coded in Section 2.2 according to the CSO database), with linear trends and annual variations in the 2000–2019 period, appear in Fig. 12. A basic summary follows:


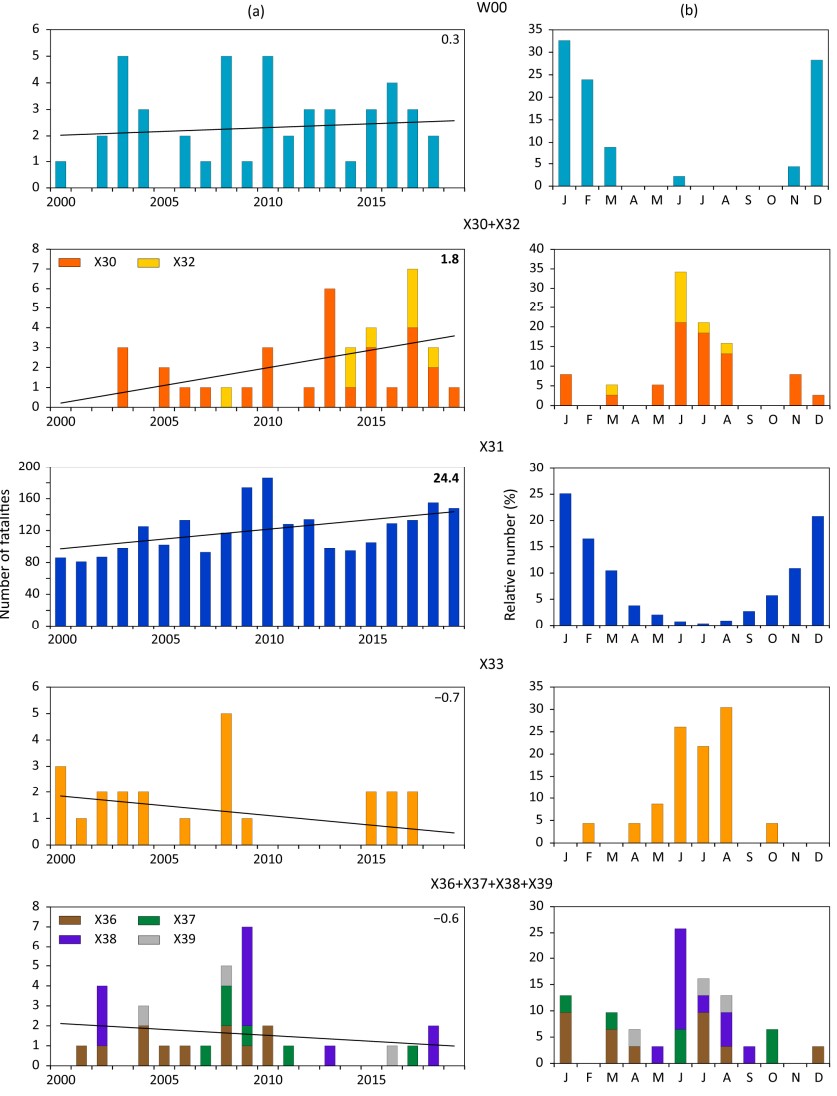

**Figure 12. Fluctuations with linear trends (a) and annual variations (b) of fatalities associated with selected types of death in the 2000–2019 period according to the Czech Statistical Office database: W00 – fall on ice or snow, X30+X32 – excessive natural heat and solar radiation, X31 – excessive natural cold, X33 – lightning strike, X36+X37+X38+X39 – avalanche, landslide or other earth movements, natural catastrophic storm, flood (inundation), and other and non-specified natural forces.**





(i) Falls on ice or snow (W00)

A total of 46 fatalities attributable to a fall on ice or snow occurred, an average of 2.3 fatalities a year. Maxima of five deaths
were recorded in 2003, 2008 and 2010; no such fatal case appeared in 2001, 2005 and 2019. The slightly rising linear trend
that emerged for 2000–2019 was not statistically significant. The great majority of fatalities (84.8 %) occurred in the three
winter months, with a maximum in January (32.6 %). In terms of sex and age, 73.9 % of these were males and 80.4 % were
elderly.

(ii) Excessive natural heat and solar radiation (X30+X32)

These groups include 38 fatalities, an average of 1.9 fatalities a year, of whom 30 (79 %) died of excessive natural heat. A
maximum of seven fatalities in 2017 (six in 2013), with none recorded in 2000–2002, 2004 and 2011, contributed to a
statistically significant increasing trend (1.8 fatalities/10 years). The summer months predominated in annual variation
(71.0 %), with a maximum in June (34.2 %). In the total number of fatalities, males (71 %) and adults (52.6 %) headed the
list, but the percentage of the elderly (44.7 %) is worthy of note.

(iii) Excessive natural cold (X31)

Excessive natural cold was responsible for the highest number of weather-related fatalities: 2407 victims, an average of
120.4 fatalities a year. The maximum appeared in 2010, with 186 fatalities, (174 in 2009), the minimum in 2001, with 81. A
rising trend of 24.4 fatalities/10 years was statistically significant. A total of 89.5 % of the fatalities were associated with the
months of the winter half-year, with a maximum in January (25.1 %). Males returned the highest percentage of fatalities
(75.5 %), then adults (67.4 %), with the elderly taking up 32.5 %.

(iv) Lightning strike (X33)

Lightning strikes were identified as the cause of death for 23 people, an average of 1.2 fatalities a year. A maximum of five
fatalities was recorded in 2008, while 2010–2014 and another four years were free of such deaths. A falling trend proved
statistically insignificant. The summer months saw 78.2 % of these fatalities, with a maximum in August (30.4 %). Most of
the deaths were categorised as males (78.3 %) and adults (91.3 %).

(v) Other natural forces (X36+X37+X38+X39)

This group includes fatalities attributable to four categories of cause of death: avalanche, landslide or other earth movement;
natural catastrophic storm; flood (inundation); and "other and non-specified natural forces". Together they constituted 31
fatalities, an average of 1.6 fatalities a year. Deaths arising out of avalanche, landslide or other earth movements on the one
hand, and flood (inundation) on the other, took up 35.5 % each. Seven people died in 2009 and no fatality occurred in six of
the years. A falling linear trend emerged, but was statistically insignificant. Fatalities were recorded in all months apart from
February and November, with a maximum in June (25.8 %). Males suffered 83.9 % of the fatalities, adults 77.4 %. Deaths
among children and the elderly were comparable (9.7 % and 12.9 % respectively).



### 4.3.2 Vehicle accidents

As reported in Section 2.3, the police yearbooks recording the accident rate in the Czech Republic facilitate the creation of series of fatalities associated with vehicle accidents in relation to weather conditions. Such conditions are classified by the police as: fog, onset of rain and light rain, rain, snowfall, rime and glaze ice, gusty winds, and other inclement weather patterns. A mean of 879.4 fatalities per year was recorded for 2000–2019, of which almost a fifth (163.2, i.e. 18.6 %) occurred during inclement weather patterns (Fig. 13a). Absolute maxima of individual weather-related events occurred in the

years 2000–2002, with minima in 2017–2019. Of the total number of 3265 fatalities in which deteriorating weather conditions were involved, rainy weather was the predominating factor (rain 35.7 %, onset of rain and light rain 25.2 %, i.e. 60.9 % altogether), followed by fog (11.1 %), snowfall (10.4 %), rime and glaze ice (8.4 %), other inclement weather patterns (7.7 %) and gusty winds (1.5 %). Considering the proportion of fatalities arising out of deteriorating weather against total fatalities in the individual years, percentages fluctuated between 27 % in 2001 and 13.3 % in 2019, while after a local

maximum in 2010 (19.9 %), this decreased steadily to a minimum in 2019 (Fig. 13b). All series of fatalities, whether associated with all deteriorating weather patterns or to their individual groups, exhibited statistically significant falling linear trends, at a significance level of 0.05, for the 2000–2019 period.

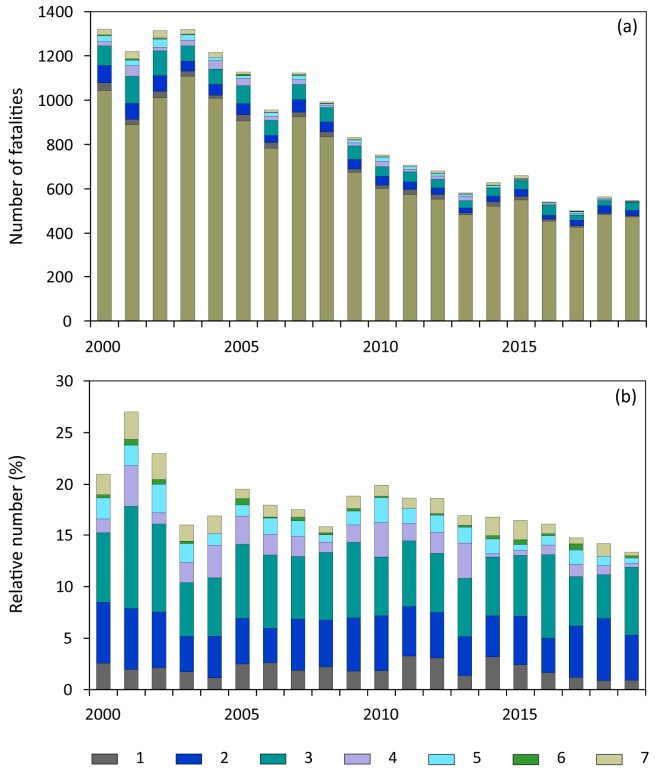

**Figure 13. Fluctuation in (a) the annual number of vehicle-accident fatalities in normal and inclement weather conditions and in (b) their relative percentage of the relevant annual number of vehicle-accident fatalities in the Czech Republic during the 2000–2019 period (1 – fog, 2 – onset of rain and light rain, 3 – rain, 4 – snowfall, 5 – rime and glaze ice, 6 – gusty wind, 7 – other inclement weather patterns).**

## 5 Discussion

### 5.1 Data uncertainty

The newly-created database of weather-related fatalities created for the purposes of this study and based on documentary data suffers from certain data uncertainties. These have been previously mentioned in, for example, contributions addressing the use of documentary data in historical climatology (Brázdil et al., 2005, 2010) and in historical hydrology (Brázdil et al., 2006). Newspaper reports have served as a vital source in the creation of databases of weather-related fatalities, to the point at which the approach has become quite common practice. They have been used, for example, for Switzerland (Hilker et al.,

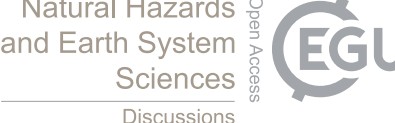

2009), Portugal (Zêzere et al., 2014), southern France (Vinet et al., 2016), Calabria in southern Italy (Aceto et al., 2017; Petrucci et al., 2018) and Mallorca (Grimalt-Gelabert et al., 2020). The results of working with Czech newspaper data over a relatively extended 20-year period may be influenced by the profound changes in society, both in the media market and in internal changes in the actual newspaper employed. The publisher and editor-in-chief of any given newspaper decide strategy. They are subject to a wide and complex range of influences: how much space will be devoted to certain kinds of

information, largely on the basis of the situation on the media market and perceived interest of target readers, and sometimes the political orientation of the news source. Further, in the background lie personnel changes, reduction of regional editorial staff (largely in the light of digitization), different quantities of space given to regional and countrywide reporting, advertising space (influencing both space for other reports and arising out of reluctance to report matters that might offend advertisers), competition in reporting, and availability of regional/local news from other bodies such as the police or the

Czech Press Agency. Reader fatigue is also important; certain kinds of fatal event became ever-more-familiar and reader interest wanes. Moreover, the real number of fatalities may also be underestimated, particularly in situations involving the severely injured being taken to hospital (e.g. after a vehicle accident, falling trees/branches, hypothermia etc.). Only seldom, and if the follow-up is deemed in some-way "remarkable", additional information is later to be found, i.e. if injured people really died. All the above circumstances may be reflected in spatial and temporal non-homogeneity of fatality data derived

from documentary evidence. It should therefore be borne in mind that the database created for the purpose of this research tends to represent a somewhat lower estimate of weather-related fatalities.

All the above serves to highlight the vital role played by critical evaluation in the use of documentary sources, especially with reference to fatality data. Inclusion in the database herein gave preference to reports containing more detailed information concerning a given fatality, particularly those that provided name (sex), age, place and the specific cause of

death. The team has remained aware of the drawbacks of employing only information summarising the total number of fatalities during any given event or period. This is also a tendency typical of the reporting of disastrous natural events, in which descriptions of material damage often take precedence over more personal matters, such as detailed descriptions of place, time and cause of fatalities. Reporting fatalities without the necessary details may result either in underestimation of real numbers on the one hand, or even exaggeration of them on the other.

Other types of bias may also appear in official databases, such as those of the CSO. Determination of cause of death on a death certificate is based on some degree of subjective perception on the part of the doctor filling it out. Even the most experienced health workers are forced to select from a broad scale of "official" definitions, in which certain categories may be understood differently by individual doctors (e.g. excessive natural heat, excessive solar radiation, and non-specified natural forces). While the database herein includes all weather-related fatalities that occurred in the Czech Republic, CSO

collects only data concerning Czech citizens, excluding the deaths of non-Czechs on Czech territory and including deaths of Czechs that take place beyond the borders. The integration and cross-checking of data between our database and that of the CSO is complicated by the fact that information about place of death has only appeared in the latter since 2010.



### 5.2 Weather-related fatalities in different databases

Weather-related fatalities in our database may be discussed in relation to those of CSO and police vehicle accident reports,
although they are not fully comparable. If frost-related fatalities (Fig. 7) are considered against "fatalities due to excessive
natural cold" according to the CSO (Fig. 11, X31), the CSO fatality figure is seven times higher, with a statistically
significant rising linear trend (in contrast to the falling and insignificant trend in our database). Both series agree upon a
maximum of fatalities in 2010 and during January and December in annual variation. While the CSO database gives
fatalities in every month of the year, no casualty was identified from May to September in our database. Both databases
show that the highest percentage of fatalities occurs among males and adults, but they differ more widely in percentages of
the elderly (32.5 % for CSO and only 10.3 % in our database, but with 21.4 % of fatalities of unknown age in the latter).

The above two fatality datasets may be compared in terms of selected characteristics of DJF severity in the Czech Republic,
as calculated from 268 homogenised temperature series. These series included mean DJF temperature, mean minimum
temperature $T_{min}$, numbers of frosty days with $T_{min} \leq -0.1$ °C, and numbers of days with $T_{min} \leq -5.0$ °C and $T_{min} \leq -10.0$ °C).
Our database exhibits a closer relationship between fatalities and temperature characteristics than that of CSO. Statistically
significant Pearson correlation coefficients lie between 0.63 (number of frosty days) and 0.87 (number of severe frost days
with $T_{min} \leq -10.0$ °C); the situation is opposite for the CSO database – between 0.53 (number of severe frost days with $T_{min} \leq -10.0$ °C) and 0.61 (number of frosty days). This reflects the higher degree of press attention paid to fatalities, particularly
among the homeless, during severe cold spells, than is the case of less extreme temperatures over the whole DJF period.
Figure 14 shows the relevant DJF correlation fields for the highest correlation coefficients in the two datasets and
fluctuations in fatalities and selected temperature characteristics in the 2000–2019 period. Extending this analysis to the
whole winter half-year, the corresponding correlation coefficients are lower than is the case for DJF.

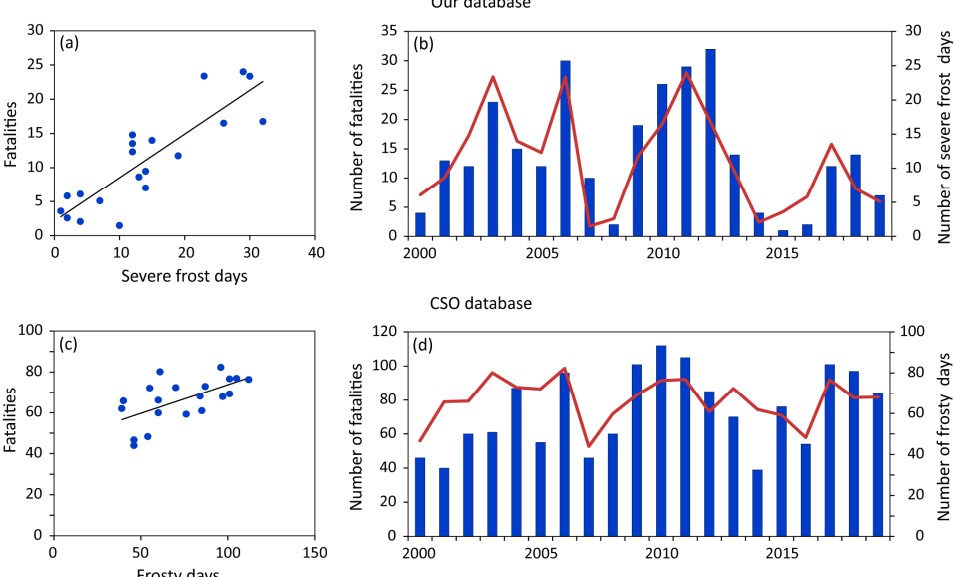

**Figure 14. Relationships between DJF frost-related fatalities and temperature characteristics for the database created herein (a, b)**
**and that of the Czech Statistical Office (c, d) in the 2000–2019 period: correlation fields of number of fatalities (a) with the number**
**of severe frost days (with $T_{min} \leq -10.0\,°C$) and (c) with the number of frosty days; fluctuations (b) in the number of fatalities**
**(columns) together with the number of severe frost days (with $T_{min} \leq -10.0\,°C$, red line) and (d) in the number of fatalities**
**(columns) together with the number of frosty days (red line).**

Although the CSO category "fall on ice or snow" (W00), with 46 fatalities, has no equivalent category in our database, nine

such fatalities appear in it. This difference can be attributed in part to fatal skiing falls on recreational slopes, which we did

not consider. Fatalities arising out of lightning strikes (X33) return slightly higher figures in CSO: 23 casualties against 15.

However, the CSO database does not make clear whether all the accidents occurred within Czech territory. For example,

reported lightning fatalities on 26 February 2003 (without location) and 14 October 2017 (Vysoká Pec given as location) are

not confirmed by any thunderstorm record from a meteorological station in the Czech Republic. From the other point of

view, there are three further fatalities dated to 2015, 2016 and 2017 in CSO that were not found in any of the print or internet

documentary sources.

On the other hand, the total of 31 fatalities from other natural forces in CSO (Fig. 12) appears to be significantly

underestimated: X36 – 11 fatalities, X37 – 6 fatalities, X38 – 11 fatalities, X39 – 3 fatalities. Our database also includes 11

fatalities (six avalanche-related and five from landslides), as reported for X36 category in CSO. However, only six fatalities

in CSO category X37 (natural catastrophic storm) do not compare well with the 32 direct casualties in "windstorms" and 14

in "strong winds during convective storms" in our database. Similarly, only 11 fatalities in the CSO X38 category (floods)





stands far lower than our 92 direct casualties during floods. The CSO X39 category (other and non-specified natural forces), does not make clear what was included in it, rendering it impossible know with which fatalities from our database it could be compared.

Heat-related fatalities are deeply underestimated in both our database and that of the CSO (X30 category) (20 and 30 fatalities respectively). The two independent series return statistically insignificant linear trends. The study herein did not give particular attention to heat-wave-related mortality, since there exist a plethora of such Czech analyses (among them Kyselý and Kříž, 2008; Kyselý and Plavcová, 2012; Knobová et al., 2014; Hanzlíková et al., 2015; Urban et al., 2017; Arsenović et al., 2019). Different data and other approaches to analysis have been utilized. Taking into account the

outstanding heat waves of 2003, 2006, 2010, 2013 and 2015–2018 from references, a noticeable peak in our database of fatalities appeared in 2006 and only two, in 2013 and 2017, in the CSO database (see Figs. 8 and 12).

    As might be anticipated, fatalities arising out of vehicle accidents relating to inclement weather conditions are underestimated quite sharply in our database (Section 4.2, Fig. 11) in comparison with those reported in the official police yearbooks (Section 4.3.2, Fig. 13): our 563 detected fatalities represent under a fifth (17.2 %) of those in police statistics.

However, the linear trends for the series in both databases exhibit statistically significant decreases. Although the categories of weather events are not the same, the differences in their corresponding percentages are very large. Compared with official police data, our database overestimates percentages heavily for glaze ice (37.5 % to 8.4 %) and snow (20.6 % to 10.4 %) and underestimates for rain (36.2 % to 35.7 % plus 25.2 % for "onset of rain and light rain" police category), fog (2.8 % to 11.1 %) and other inclement weather (2.8 % to 9.2 %). The above discrepancies in our data compared with police data may

be explained by the nature of public information concerning vehicle accidents. The greater part of vehicle accidents reported in the press only seldom provide details of ambient weather patterns, perhaps in the event of snow or glaze-ice calamities, or when major highways are closed by weather (and consequent accidents). From reports of the type that "driver did not adapt speed to road conditions" or "the reason for the accident is subject to further investigation", it is impossible to derive weather information. Similar interpretation difficulties exist for vehicle accidents described as "skid on slippery road", which we

interpreted as glaze ice during the winter months. Of course, not every vehicle accident with casualties was reported in the press, although the total numbers of fatalities generated by an extreme event, or at any particular weekend, were often mentioned without necessary details.

    Despite all these uncertainties, our database contrasts with other, official, databases in providing more detailed information about the circumstances surrounding fatalities and permits a more complete overview of the causes and consequences of

fatal events.

### 5.3 The broader context

Beyond the work herein, in which our figures for weather-related fatalities are considered against the databases kept by the CSO and the Czech police, there is a shortage of similar papers for direct comparison of results. For the Czech Republic, Daňhelka (2018), reported the creation of the Czech Hydrometeorological Institute database of disastrous historical


phenomena and their impacts from 1993 onwards. This provided figures of 235 fatalities that broke down into 126 males, 47
females and 62 without sex specification. In particular, he mentioned 96 fatalities for the 2005–2015 period (42 – floods, 21
– flash floods, 16 – frost and snow, 9 – windstorms, 6 – lightning and avalanches, 2 – landslides), a gross underestimate
considered against the 406 fatalities for these six categories in our database for the same period. A very preliminary database
of weather-related fatalities in the Czech Republic that appears in Brázdil et al. (2019b) included only 181 fatalities for
2000–2018, as against the 1145 casualties herein. The numbers of flood-related fatalities in the Czech Republic mentioned
by Brázdová (2012), 56 for 2000–2010, and 65 for 2000–2013 reported by Punčochář (2015), are significantly lower than
those in our database (81 and 103 fatalities respectively). The European Severe Weather Database, maintained by the
European Severe Storms Laboratory, has a record of weather-related fatalities in Europe covering 1981 onwards (Dotzek et
al., 2009). It reports 33 such fatalities in the Czech Republic for 2003–2018. The majority of fatal events were attributable to
floods (60 %), followed by strong winds (27 %) and lightning strikes (9 %).

In other European countries, Badoux et al. (2016), addressing the 1946–2015 period for Switzerland, found an average of
14.7 fatalities a year. Terrain that includes the Alps inevitably dictated that the highest percentage of fatalities was
attributable to snow avalanches (37 %). This was followed by lightning strikes (16 %), floods (12 %), windstorms (10 %),
rockfalls (8 %), landslides (7 %) and "other processes" (9 %). Antonescu and Cărbunaru (2018) recorded 724 lightning
fatalities between 1999 and 2015 in Romania, an average of 42.6 fatalities a year); they identified males aged 10–39 years in
rural areas as the most vulnerable group. More recently, Špitalar et al. (2020) reported 74 flood fatalities resulting from 10
floods in Slovenia between 1926 and 2014.

Some of the data concerning flood-related fatalities from our Czech database was integrated, together with that from eight
other regions, mainly Mediterranean, into the EUFF database for an analysis of the 1980–2018 period, performed by Petrucci
et al. (2019a). While fatality series for Greece, Italy and southern France indicated increasing trends, the opposite was
evident for Turkey and Catalonia (Spain). The remaining regions – Portugal, the Balearic Islands, Israel and the Czech
Republic – exhibited quite stable linear trends. In more detail, the structure of their total of 2466 flood fatalities detected
features similar to those disclosed in our analysis (see Fig. 1): a prevalence of male fatalities aged 30–49 years, the majority
of deaths outdoors, drowning as a primary cause of death, followed by indirect deaths arising out of heart failure. Casualties
were most frequent in vehicles carried away by water or mud. Paprotny et al. (2018), in an analysis of floods that did damage
in 37 European countries from 1870 onwards, found a substantial decrease in flood fatalities, despite increases in annually
inundated areas and the numbers of people affected.

Sharma et al. (2020) investigated over 4000 winter drowning events resulting from falling through ice in the course of a
range of activities, covering 10 Northern Hemisphere countries. Children and adults aged up to 39 years were at the highest
risk. They maintained that the potential for this type of accident was rising with warmer winters. Our database documents
only 14 such fatalities during 2000–2019, occurring especially when crossing or skating on insufficiently frozen water
bodies of natural or anthropogenic origin or as a result of hazardous behaviour on the part of children (29 %), adults (36 %)
or the elderly (21 %). Age was not specified for two of the casualties.



Summarising the results of trend analysis for weather-related fatalities in the Czech Republic during the 2000–2019 period,
statistically significant falling linear trends were revealed by both methods of linear trend calculation for convective storms,
glaze ice, snow and all weather-related fatalities, as well as for windstorms according to the Mann-Kendall test. The
remaining groups of events returned statistically insignificant trends. In the CSO database, statistically significant rising
fatality trends emerged for categories X31 (excessive natural cold) and X30+X32 (excessive natural heat and solar radiation)
according to both methods applied. The trend lines appearing in Figs. 1–13 show lower values of slope for Sen's estimate,
since this approach is less sensitive to outliers, providing reliable results even in situations where one extremely high value
may influence the slope of the trend line in the classical least-square regression method (Gilbert, 1987). In this sense, our
estimate of significant trends provides quite consistent results. These trends may be compared with those in other
contributions. For example, Analitis et al. (2008), addressing 15 European cities in 1990–2000, maintain that a 1 °C decrease
in October–March temperatures contributed to a 1.35 % increase in the daily number of total natural deaths, with comparable
or higher increases in cerebrovascular (1.25 %), cardiovascular (1.72 %) and respiratory (3.30 %) deaths. Plavcová and
Urban (2020), using mortality data for the Czech Republic for 1982–2017, recorded that sudden rises in minimum
temperature and drops in pressure had a generally significant impact on excess mortality, by 3.7 % and 1.4 % respectively.
This impact was significantly exacerbated if the two events occurred simultaneously or when they were compounded by
other extremes (heavy precipitation, snowfall, maximum temperature rise) and combinations thereof (14.4 %). Holle (2016)
reported a large reduction in lightning fatality rates for western Europe and some other regions during recent years,
associating it with changes in society from the largely rural, agricultural to the primarily urban. Franzke and Torelló i
Sentelles (2020), collating figures worldwide, found statistically significant increasing trends for heat-wave- and flood-
related fatalities.

Statistically significant falling trends also appear in the numbers of Czech fatalities associated with vehicle accidents taking
place in bad weather conditions such as rain, snow, glaze ice, fog, etc. if conclusions are based on the authors' database and
that of the police in 2000–2019. Because the majority of these cases take place during the winter half-year, in which warmer
winters with "better" weather for traffic may play some role, the influence of other factors may be higher. These could
include such matters as safer cars, better roads and their closer maintenance, weather forecasting, general media warnings
before particularly adverse weather, raising of public awareness of road safety, and improved emergency services and health
systems, among other things. Trends in the Czech Republic are partly, for example, in agreement with Andrey (2010), who
disclosed a downward trend in the relative risk of casualty during rainfall and no significant change during snowfall in an
analysis of 1984–2002 data concerning weather-related crash risk in automobile transport in 10 Canadian cities.

The question remains open as to the extent to which variability and trends in weather-related fatalities may be attributed to
specific factors in the light of the data uncertainty discussed in Section 5.1. Climate variability in recent decades, as
documented by many climatological papers (see e.g. Brázdil et al., 2017, 2019a, 2020; Zahradníček et al., 2020), and socio-
economic factors, as mentioned for weather-related car accidents in the previous paragraph, all have parts to play. Each
individual fatality is a synergy of different, largely random circumstances, including hazardous behaviour in many cases.





Although Franzke and Torelló i Sentelles (2020) found that socio-economic factors had no significant direct impact on statistically significant increasing trends in heat-wave- and flood-related fatalities worldwide, and argued for the significant influence of climate variability on the numbers of fatalities, the authors maintain that, at regional or national scales, socio-economic factors and hazardous behaviour may be of high importance. Of course, this in no way implies that the real and pressing effects of climate variability, as shown through the example of Fig. 14, should be overlooked.

## 6 Conclusion

The following conclusions may be drawn from this analysis of weather-related fatalities in the territory of the Czech Republic during the 2000–2019 period:

(i) The weather-related fatality database for the Czech Republic in the 2000–2019 period, derived from the daily newspaper *Právo* and its internet counterpart *Novinky.cz*, partly supplemented by further documentary evidence, constitutes a unique data source for the study of the spatiotemporal variability and structures of such casualties.

(ii) Weather-related fatalities in the database herein may be attributed in particular to frost (cold spells), glaze ice, rain and snow. In annual distribution of fatalities, the winter months are predominant. The falling linear trends evident in the number of fatalities in 2000–2019 were statistically significant only for those arising out of all weather factors together and for casualties arising out of convective storms, snow and glaze ice; linear trends in flood- and windstorm-related fatalities proved insignificant.

(iii) The structure of weather-related fatalities indicates the highest percentages for males, adults, indirect types of death, vehicle accidents due to inclement weather conditions, and freezing or hypothermia, night or morning times of deaths and hazardous behaviour on the part of casualties or other persons responsible for the fatal incident.

(iv) Fatalities derived from the database of Czech Statistical Office, ordered according to the type of weather-related death, exhibit statistically significant rising trends in the "excessive natural cold" and in the combined "excessive natural heat and solar radiation" categories. In terms of sex and age structures, males and adults predominate.

(v) Vehicle-accident fatalities during bad weather conditions, as extracted from police yearbooks, show statistically significant falling trends for all and individual weather events. Rainy weather, with *c.* 61 %, predominates in the latter category, followed by fog, snowfall and glaze ice.

(vi) Trends in weather-related fatalities in the Czech Republic reflect, in addition to recent weather and climate changes, certain socio-economic factors and especially people's behaviour, which may be for *c.* 75 % characterised as hazardous.

However, existing data uncertainties in the evaluation of recognised trends should always be borne in mind.

**Appendix A**

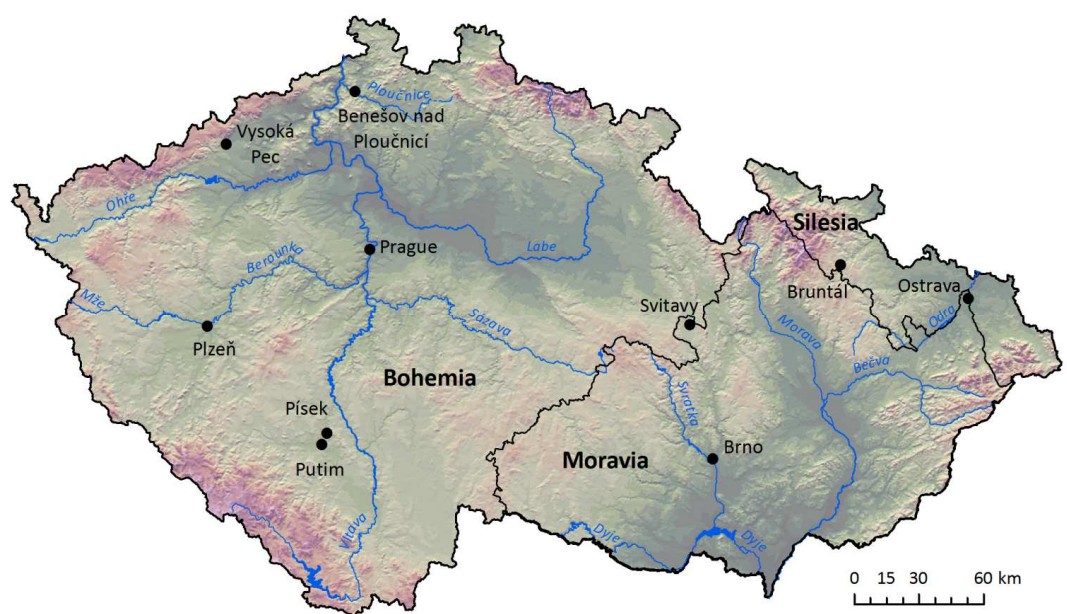

**Figure A1. Locations in the Czech Republic mentioned in this paper.**

**Data availability.** Fatality data from authors database can be made available by the authors upon request. Fatality data of the Czech Statistical Office and of the Police of the Czech Republic can be found in the related published yearbooks (CSO, 2020a, 2020b; PCR, 2020).

   **Author contributions.** RB designed and wrote the paper with contributions from all co-authors. KC created fatality database
and made basic analyses with data, including finalising of figures. RB, KC, LD, JŘ and LŘ extracted and collected data from newspapers. PZ contributed with analysis of frost-related fatalities and temperature characteristics. PD evaluated linear trends in fatalities and their significance.

   **Competing interests.** The authors declare that they have no conflict of interests.


   **Acknowledgements.** This study was financially supported by the Ministry of Education, Youth and Sports of the Czech Republic for the SustES – Adaptation strategies for sustainable ecosystem services and food security under adverse



environmental conditions, project ref. CZ.02.1.01/0.0/0.0/16_019/0000797. Jan Řehoř was supported by Masaryk University within project ref. MUNI/A/1356/2019. The Czech Statistical Office, Prague (Jana Audy, Robert Šanda) is acknowledged
for providing us with selected data of fatalities for 2000–2019. The regional headquarters of the South Moravian Regional Police (Jindřich Rybka) is acknowledged for providing data concerning fatalities arising out of vehicle accidents for 2000–2006. Tony Long (Carsphairn, Scotland) helped work up the English.

**Financial support.** This research has been supported by the Ministry of Education, Youth and Sports of the Czech Republic
for the SustES – Adaptation strategies for sustainable ecosystem services and food security under adverse environmental conditions, project ref. CZ.02.1.01/0.0/0.0/16_019/0000797.

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
