# Peer review of "Fatalities associated with the severe weather conditions in the Czech Republic, 2000–2019"

_Natural Hazards and Earth System Sciences, 2021_

## Referee Comment (RC1)

**Review of manuscript NHESS-2021-14:**

**"Fatalities associated with the weather in the Czech Republic, 2000–2019"**

**by Rudolf Brázdil, Kateřina Chromá, et al.**

**1) General comments**

Dear Editor,

dear Authors

This contribution of Brázdil et al. presents an analysis of weather-related fatalities in the Czech Republic during the period of 2000–2019. The study was very meticulously conducted and the manuscript is well organized. The manuscript is moderately well written and largely quite easy to understand. However, and as far as I can judge as a non-native English speaker, the sentence structure and punctuation is sometimes a bit special/awkward. This is a bit confusing here and there. Maybe a final proof-reading by a native English speaker could be helpful. The overall text seems a bit long to me. I make suggestions in various places on how it could be shortened slightly (if this is desired by the editors).

Presumably, data collection was very laborious, which makes this work all the more valuable. The authors have also given a lot of thought to the completeness of their data, which is also important, and discuss this point in great detail in the discussion. Furthermore, the authors show how difficult it is to compare, let alone combine, different data sources (e.g., national statistical offices or police). More attention should be paid to this point in the future. It is difficult enough for a country or region to compile databases on fatalities caused by natural hazards and weather extremes, useful synergies would be a great help.

In Europe, several similar overviews o those of the authors have emerged in recent years. The effort of the authors Brázdil et al. fits in perfectly here. This study is very important, not the least to help Czech authorities to better identify potential improvements in hazard prevention related to severe weather situations and natural hazard processes and to reduce the number of victims in the future.

I have compiled my criticisms and comments in quite a bit of detail below and refer you to the second part of this review. Among other things, the title could be worded a bit more "crisply". Here I just want to briefly mention my clear main point of criticism, which is explained more in detail in the comments to the Methods and Discussion sections. For me, the division of the various fatalities considered here into nine classes/types is problematic. Meteorological causes and hydrological (as well as geomorphological and avalanche specific)

consequences are mixed. In my opinion, this needs to be reconsidered or at least better and more extensively argued in the text.

Moreover, the importance of traffic accidents in this compilation is immense. Because (at least it seems so to the reader) as soon as the weather conditions were not optimal during an accident on the road, such a fatal accident was recorded. Here, in my opinion, the authors need to better explain how they proceeded and why.

In summary, this manuscript will definitely be of interest for the research community and therefore should in my opinion be published in NHESS. I am looking forward to it. Given the considerable number of minor comments, and the general criticism with a "major" character, I suggest that the paper be accepted pending moderate revisions.

I provide below a list of comments and suggestions specific to the different sections of the article. I ask the authors to consider these.

**2) Specific comments regarding the different sections**

**Title**

The title is a bit general and unspecific (weak). I suggest something slightly more specific, such as "Fatalities associated with severe weather in the Czech Republic, 2000–2019" or "Fatalities associated with severe weather situations in the Czech Republic, 2000–2019". In the second suggestion, "situations" could also be replaced by "events".

**Introduction**

The introduction of the article has an adequate length and it draws the reader's attention to the interesting and important topic of this article. In some places, the wording could be a little more precise, and one aspect that is important for the article has not yet been taken into account enough (see comments below).

Also, , the authors do not "sell" their manuscript very well in the last paragraph of the introduction. Instead of describing the structure of the text with its different sections (which is fine but not absolutely necessary), they should try to better explain the goals of this research. Moreover, it is also important for the readers to know what is new in this paper, respectively what progress the study represents for disaster risk reduction in the Czech Republic specifically and in Europe in general.

*Further comments:*

L33: The authors mention world-wide or continental scales; but there are also many studies and articles that examine fatalities due to weather events or natural hazards at the national level. Is there a reason why these are not mentioned here?

L34: Consider changing to "presented a broad world-wide" (instead of "broader")

L40-41: This is a little confusing. The authors write that "particular attention has been paid to deaths associated with heat-waves and floods". Where? In this article specifically or is this meant in general?
The focus within this introduction on heat waves and floods should be better described and justified. After all, almost half of the fatalities described in this study are due to traffic accidents in problematic weather situations (ice and snow covered roads, fog and rain). Why is this not addressed in the Introduction? Are there no compilations and studies on this kind of incidents in research?

L53: Please reconsider the formulation "although other countries have their shares"

L54: The article by Hilker et al. (2009) seems to primarily focus on financial damage caused by floods, landslides and other processes. I guess it is by all means citable here but for Switzerland, Badoux et al. (2016) would probably fit a bit better because it focusses on natural hazard fatalities.

Line 54&57: The article by Petrucci et al. you cite first in the course of the text should be labelled 2019a (and not 2019b)

L62-63: The Salvador et al. (2020) study should rather be mentioned in the second paragraph of the Introduction where studies on heat waves and droughts are addressed. It there a special reason it is placed here?

L74-75: When the authors mention "The general increase in climatic and weather hazards…" do they mean an increase in the number of events? Please include a literature citation that supports this statement.

L78: "The work is based on its own mortality database…" is a bit an awkward formulation. Consider making the text clearer, for example like this: "The work is based on the mortality database compiled by the authors from newspaper data and other official/administrative sources of information"

L89-81: Consider changing to "The results in Section 4 describe weather-related fatalities for various weather phenomena and for all phenomena combined". Moreover, the second part of the sentence on line 80 does not match the first grammatically.

**Data**

The three examples of fatality records (L99-116) could be moved to the appendix to shorten the text.

*Further comments:*

L99: Change to "One of the fatalities of the disastrous August 2002 flood was…"

L133: What are "normal weather conditions"? This should be explained. I guess the authors mean dry weather with good visibility etc. I am not sure if "normal" is the good term, because would mean "rain" is generally an abnormal phenomenon (which is not true).

**Methods**

The typification of weather events to distinguish weather-related fatalities is not consistent. This is actually my most important criticism of this otherwise very carefully and precisely edited project/article. In my opinion, there is a big confusion between (1) weather phenomena, (2) resulting hydrological and geomorphological processes, (3) resulting traffic accidents, and (4) resulting other accidents.

If floods are considered, why, for example, are mass movements such as landslides, rockfall, and debris flows not considered (at least not mentioned)? These can also be triggered by precipitation. The inconsistency of the present approach is also shown by the fact that avalanches (a mass movement) are taken into account within the weather type "snow", but the above-mentioned processes (various landslide types) do not seem to have been considered anywhere.

Furthermore, considering road accidents in ice, snow, rain and fog is problematic in my opinion. I am familiar with e.g. the studies by Diakakis & Deligiannakis (2013, Vehicle-related flood fatalities in Greece) or by Coates (1999, Flood Fatalities in Australia, 1788-1996). Both publications show that people often die by drowning when they act carelessly in cars during floods and, for example, enter flood areas / plains or try to cross a watercourse during floods, be it over a bridge or by using a ford. These are flood victims in my opinion. You can also call them traffic victims, but they died mainly because of a natural hazard. But if a driver does not adapt his driving style to the external circumstances (on a wet road, snow-covered road, icy road or in poor visibility) and is involved in a crash, that is for me first and foremost a traffic accident. Of course, the "bad" weather plays a role and the victim can be classified as a weather-related fatality, but a clear distinction must be made from, for example, lightning fatalities, flood fatalities, etc., and the topic should be discussed in detail.

*Further comments:*

L147: Regarding the "locality": I suppose this is the locality of the accident or event and not the actual place of death (which can also be in the hospital, for example). Perhaps this could be added.

L155-158: Why do the authors not use one of the three examples described in section 2.1 (L99-116); that would make sense in my opinion.

L160: Maybe indicate that (in contrast to flash flood) these are triggered by long-lasting rainfall; thus consider "single-day or multi-day persistent rainfall" or something similar

L161: I would use the term "snow-melt flood" instead of "snow flood"

L162: Change to "in the rivers"

L162-163: Here is a typical case where it is difficult to distinguish the categories (see general comment on methodology): How are fatalities from the category "floods / flash floods" and "convective storm" distinguished? Both are triggered by very intense, short-term precipitation events. The former category is the consequence, the latter category is the cause. This does not work for me.

L164: To distinguish this category (ii) from the next (iii), I propose to call them "Non-convective windstorms" or "(Non-convective) windstorms". Maybe even: "Non-convective, non-tornadic windstorms"

L167: Downpour (short and very intense rainfall) and hail are phenomena that potentially trigger overland flow and subsequent inundations as well as floods in water courses, overflow and subsequent inundations (in mountainous regions intense rainfall can also trigger e.g. channelized debris-flows); as I understand it, these process chains are covered in (i). What I don't understand exactly is whether there were people in the Czech Republic who were killed directly by heavy precipitation (very unlikely) or hail (possible but unlikely).

L168: Similar to the wind, I would call this category (if it remains so) "Non-convective rain".

L168: Same here (as comment on line 167): were fatalities in this category or event/phenomena type killed by rain? I already mentioned it above in my general comments on the Methods (and will address it again below regarding lines 552-567): I am not sure whether a traffic accident that occurs during rainfall should (always or per definition) be included in the analysis of weather-related climate fatalities. If a person drowns in their car during a flood, it's a clear-cut case. But if a person does not maneuver properly in wet road conditions, it is not a clear case to me. After all, driving on wet roads is a very common thing. The actual cause of an accident can then also be a completely different one.

**Results**

Section 4 (Results) is extremely detailed and thorough with its nine subsections presenting the various hazardous weather phenomena. The question arises whether all figures 1 to 9 belong in the actual article. Alternatively, Figure 10 (which covers all weather phenomena) could be printed on a full page. The points on the map could be colored according to their type (or different symbols could be used). For the interested reader, Figures 1 through 9 would then be placed in a supplementary material appendix. They contain a great deal of important information, but lengthen the text massively. However, this is the decision of the associated editor or the editor-in-chief. Both options are actually OK for me

*Further comments:*

L191: I'm not sure if "events" is quite correct in this subtitle; perhaps "categories" or "types" would be more appropriate

L195: Change to "heavy flood in August"
Maybe the authors can make a reference to an article or report which describes the event?

L197: Change to "In terms of monthly distribution,"

L210: Instead of "type of death" I would rather write "type of fatality" here in the caption, exactly the same as on line 153. Please check the entire document and use consistent term.

L210: Instead of "Symbols", consider using "Abbreviations"

L237: As already mentioned at line 167, I do not doubt that hail can kill people, but this number (8) seems to be quite high.
In the meantime, however, I think I have understood: these are probably road accidents, right? If so, that would have to be indicated here absolutely, otherwise this leads to misunderstandings).

L238: I suggest changing to something like "were simply indicated as having occurred "during a thunderstorm"

L249-250: Change to "4 – rain or hail", same as on line 237-238 (please be consistent);
also, "3 – thunderstorm" is confusing, change to something like "3 – unclear, occurred during thunderstorm"

L253: As mentioned above (Methods) I would use "4.1.4 Non-convective rain" as a subtitle

L256: Change to "In terms of monthly distribution,"

L259: It is not clear to me what the authors mean by "smaller regions" of the Czech Republic. Try to describe more clearly here.

L260-261: "All these fatalities were classified as indirect consequences of vehicle accidents." I do not understand this statement. To me, these fatalities appear to be a direct consequence of a vehicle accident and an indirect consequence of a non-convective rain event.

L275: With "smaller areas", do the authors mean "clusters"?

L294: Change to "and 9 % occurred on"

Figure 10: Using a color code in the bar charts is actually a good idea, but some of the colored sections are just too small or narrow to be recognized. Maybe an alternative way of displaying the proportions of the different weather types could be found (and placed e.g. in the appendix).

L381: I would delete "(56.1 % altogether)", it is not really necessary

L382: "Nearly a third of them (30.3 %) fell victim to"; this is confusing. Do the authors mean nearly a third of the 66 or of the 13?

L387-388: The end of this list is a bit confusing.

L393-394: I do not understand this sentence:
"Using the vehicle accident casualties classified within "indirect deaths" and "hazardous behaviour" (96.8 %), 94 % of them died on roads and the remaining 6 % in built-up areas and the countryside." Please try to clarify this.

L405: Consider changing subtitle to: "4.3.1 Official demographic databases (CSO)"

L444: Consider changing subtitle to: "4.3.2 Police database of vehicle accidents"

L448: For clarity, consider extending to: "A mean of 879.4 total fatalities per year was recorded for 2000–2019 due to traffic accidents, of which"

L450: Is "deteriorating" really the right term here? In the caption of Fig. 13 you use "inclement". Consistency is important

L459-462: Please state clearly in the caption which data is shown here (own database or Czech police database). This is especially important for those readers who first look at all the figures before (perhaps) reading text.
Also, what is "normal" (weather); where is the threshold between normal and not normal (inclement); this needs to be addressed. I also mention this point in my comment to the discussion.

Figure 13: Which color shows "normal" weather in Figure 13? Have you declared that?

**Discussion**

The **first subsection** of the discussion (5.1 Data uncertainty) is well structured and very detailed (perhaps almost a bit too long/detailed). It covers the important points that control data uncertainty. The part between lines 473 and 481 could possibly be streamlined a bit.

The reviewer would be very interested in one additional point. The study was made (mainly) by means of information from the print edition of one daily newspaper (Právo) and its

Internet counterpart (Novinky.cz). The use of different editions devoting space to individual regions of the Czech Republic ensured that decentralized events could also be identified. But to what extent? I would like the authors' assessment of whether the study would have benefited from (i) examining several daily newspapers and/or (ii) including regional and local newspapers. It is clear to me that this endeavor would probably have been far too burdensome, but I think this question should be briefly discussed in this section of the manuscript.

In the **second subsection** of the discussion (5.2 Weather-related fatalities in diff. databases) the authors mention the most important point at the beginning on line 505: the different data sources "are not (fully) comparable". The reviewer would even say they are not (really) comparable. (In all honesty,) This somewhat reduces my interest in this part of the study (or discussion). Nevertheless, it is of course important that the authors point out the problems with the comparative use of data from different sources. And they do it extensively and completely. If the editor concludes that the paper needs to be shortened somewhat, it could be amended here. Figure 14 and its description in the text, while interesting, takes up quite a bit of space. A more concise approach could certainly be taken here.

The **third subchapter** of the discussion (5.3 The broader context) is, in my opinion, a bit long and, above all, not very well organized (with the exception of the very well worded and meaningful last paragraph, L 638-648). I do not find a common thread in it. I feel that the authors list a bit too many comparisons with various other studies (Czech an and international) in this subsection. The order in which they are mentioned does not always seem clear and partly I am not sure that all comparisons are very helpful in assessing the facts and figures in this article.

*Further comments:*

L465: Avoid using "created" twice in the same sentence. Maybe change to: "The new database of weather-related fatalities developed for the purposes of this study…"

L471-472: Consider changing to "over a 20-year period may". For, yes, societal changes took place during these 20 years, but this time period does not represent a very long time period compared to study periods in other similar investigations such as 200 plus years in Coates (1999) or 50 years in Atkins and Williams (2000).

L506: The reference should be made to (Fig. 12, X31) and not to Fig. 11

L510: In "These series included mean DJF temperature, mean minimum temperature Tmin,…" are mean *daily* temperatures and mean *daily* minimum temperatures meant?

L531: consider extending to "23 casualties against 15 in our data collection."

L533-534: I do not know the spatial distribution and density of meteorological stations in the Czech Republic (the values are probably comparable with the values in other European countries), but since convective thunderstorms can occur on a very small scale, this argument is not necessarily a strong one.

L534-535: I do not understand "From the other point of view". Why "other", I am a bit confused. The 26.02.2003 and 14.10.2017 incidents come from the CSO database, right?

L539: This is the first time in the text that landslides are mentioned. Up to this point, it is not clear to the reader that landslides are considered in this compilation. This would have to be adjusted in chapter 3.

552-567: The statements made by the authors in this paragraph confirm my concern and reluctance to include (all) traffic accidents in the compilation of an overview of weather-related fatalities. This is because, in a sense, it is a question of "thresholds." Take, for example, the category "Rain" in the present database, or "Rain" and "Onset of rain and light rain" in the police data. With an annual precipitation total of 450 to 550 mm in the Prague region, a light precipitation of 5 mm in say 4 hours cannot be considered as "abnormal" weather (there are probably about 80-100 rainy days in Prague annually).  A traffic accident occurring in such conditions cannot, in my opinion, result in a weather-related fatality.
Theoretically, only cases that occur as a result of exceptional weather should be taken into account. Or how do the authors see it? I think this needs to be addressed in this article. Even more so when you consider that 563 of the 1164 deaths (48.4%) in this database are traffic accidents. I dare to say that this percentage is quite extraordinary compared to countries worldwide (I don't think the authors mention this in section 5.3, for example).

L586-589: I do not think these two studies are comparable. The present work lists weather-related fatalities. The Swiss study, on the other hand, describes natural hazard processes and does e.g. not include frost and heat deaths, nor does it include traffic accidents that occur during "non-normal" weather conditions.

L590: Typo, please change to "an average of 42.6 fatalities a year;" (without closing parenthesis)

L597: What does "their" (in "the structure of their total of") refer to? The entire EUFF or the "remaining regions"? Please try to be accurate here.

L603-608: Since the article is already quite long and this type of accident occurs only slightly in the previous study, I would omit this section. The number 4000 from Sharma et al (2020) is actually worthless if no time period can be associated with it.

L611: Consider clarifying and changing to: "glaze ice, snow and for all weather-related fatalities. Windstorms show a significant decreasing linear trend only according to the

Mann-Kendall test. The remaining types of weather-related fatalities returned statistically insignificant trends."

L619: I don't understand exactly what the authors mean by "total natural deaths." That has nothing to do with weather-related deaths, does it? If so, why is that of interest here? I propose to significantly shorten the text from line 617 to 624.

L629-637: It should be taken into account that, in general, the number of traffic fatalities is decreasing in many European countries, regardless of the weather. This development is very clear, e.g. in Germany and France, where the number of traffic fatalities has decreased 5 to 6 times since the early 1970s. Many of the reasons for this are given by the authors in the text (L 633-635). The citation of the Audrey (2010) study is interesting, but this general trend should definitely be stated in this paragraph, for example at the end of it.

L631-632: I do not understand the use of "Because" at the beginning of the sentence. I suggest rephrasing (maybe making two sentences). The second sentence could start with "However, the influence..."

**Conclusions**

The conclusions are concise, which I like, and present the key findings of the study. They can be read and understood independently of the overall text, which is becoming increasingly important for the large number of so-called "cross-readers" these days.

I would add in point (iv), (v), or directly below these points that comparing fatality data from different sources within the same country is problematic or difficult (as shown in Sections 5.2 and 5.3) and that comparison with similar studies from other countries is also challenging (as shown in Section 5.3).

*Further comments:*

L651: Consider changing to "for the Czech Republic in the period 2000–2019,"

L653: Please check "constitutes a unique data source for.. ...structures of such casualties". What do the authors mean exactly by structures of such casualties? Clarify.

L655: Change to "In the monthly distribution of fatalities,"

L6655-656: Consider changing to "The decreasing linear trends in fatalities from 2000-2019 were statistically significant only for fatalities caused by all weather factors combined and for fatalities caused by..."

L659: Again, I have a Problem with the term "structure of fatalities"; but maybe the problem is with me. Do the authors mean (?):

"The composition of weather-related fatalities  with respect to different distinguishing criteria indicates…"

**References**

Atkins, D. and Williams, K. (2000): 50 Years of Avalanche Deaths in the United States . In: Proceedings of the 2000 International Snow Science Workshop, October 1-6, Big Sky, Montana, p. 16-20 [https://arc.lib.montana.edu/snow-science/objects/issw-2000-016-020.pdf]

Coates, L. (1999): Flood Fatalities in Australia, 1788–1996, Australian Geographer, 30, 391–408 [doi:10.1080/00049189993657]

Diakakis, M. and Deligiannakis, G. (2013): Vehicle-related flood fatalities in Greece, Environmental Hazards, 12, 278–290 [doi:10.1080/17477891.2013.832651]

---

## Author Response (AR1)

**RESPONSES TO REFEREE 1**

**Review of manuscript NHESS-2021-14:**
**"Fatalities associated with the weather in the Czech Republic, 2000–2019"**
**by Rudolf Brázdil, Kateřina Chromá, et al.**

**1) General comments**

Dear Editor, dear Authors
This contribution of Brázdil et al. presents an analysis of weather-related fatalities in the Czech Republic during the period of 2000–2019. The study was very meticulously conducted and the manuscript is well organized. The manuscript is moderately well written and largely quite easy to understand. However, and as far as I can judge as a non-native English speaker, the sentence structure and punctuation is sometimes a bit special/awkward. This is a bit confusing here and there. Maybe a final proof-reading by a native English speaker could be helpful. The overall text seems a bit long to me. I make suggestions in various places on how it could be shortened slightly (if this is desired by the editors).

Presumably, data collection was very laborious, which makes this work all the more valuable. The authors have also given a lot of thought to the completeness of their data, which is also important, and discuss this point in great detail in the discussion. Furthermore, the authors show how difficult it is to compare, let alone combine, different data sources (e.g., national statistical offices or police). More attention should be paid to this point in the future. It is difficult enough for a country or region to compile databases on fatalities caused by natural hazards and weather extremes, useful synergies would be a great help.

In Europe, several similar overviews o those of the authors have emerged in recent years. The effort of the authors Brázdil et al. fits in perfectly here. This study is very important, not the least to help Czech authorities to better identify potential improvements in hazard prevention related to severe weather situations and natural hazard processes and to reduce the number of victims in the future.

I have compiled my criticisms and comments in quite a bit of detail below and refer you to the second part of this review. Among other things, the title could be worded a bit more "crisply". Here I just want to briefly mention my clear main point of criticism, which is explained more in detail in the comments to the Methods and Discussion sections. For me, the division of the various fatalities considered here into nine classes/types is problematic. Meteorological causes and hydrological (as well as geomorphological and avalanche specific) consequences are mixed. In my opinion, this needs to be reconsidered or at least better and more extensively argued in the text.

Moreover, the importance of traffic accidents in this compilation is immense. Because (at least it seems so to the reader) as soon as the weather conditions were not optimal during an accident on the road, such a fatal accident was recorded. Here, in my opinion, the authors need to better explain how they proceeded and why.

In summary, this manuscript will definitely be of interest for the research community and therefore should in my opinion be published in NHESS. I am looking forward to it. Given the considerable number of minor comments, and the general criticism with a "major" character, I suggest that the paper be accepted pending moderate revisions.

I provide below a list of comments and suggestions specific to the different sections of the article. I ask the authors to consider these.

RESPONSE: We would like to thank the reviewer for a careful evaluation of our paper and summarizing of general comments which we are trying to explain below. Because this study represents the first basic paper with attention to fatalities related to weather in the Czech Republic in this complex view, we see as important to explain different aspects of this topic. We believe that there will be not necessary to reduce the extent of this paper which can became a core paper for further studies with this and similar orientation in the Czech Republic. Concerning of English, we would like to only add that the manuscript was corrected by a native, Mr. Tony Long.

**2) Specific comments regarding the different sections**

**Title**

The title is a bit general and unspecific (weak). I suggest something slightly more specific, such as "Fatalities associated with severe weather in the Czech Republic, 2000–2019" or "Fatalities associated with severe weather situations in the Czech Republic, 2000–2019". In the second suggestion, "situations" could also be replaced by "events".
RESPONSE: Accepted, we changed a title as follows: "Fatalities associated with the severe weather conditions in the Czech Republic, 2000–2019."

**Introduction**

The introduction of the article has an adequate length and it draws the reader's attention to the interesting and important topic of this article. In some places, the wording could be a little more precise, and one aspect that is important for the article has not yet been taken into account enough (see comments below).

Also, the authors do not "sell" their manuscript very well in the last paragraph of the introduction. Instead of describing the structure of the text with its different sections (which is fine but not absolutely necessary), they should try to better explain the goals of this research. Moreover, it is also important for the readers to know what is new in this paper, respectively what progress the study represents for disaster risk reduction in the Czech Republic specifically and in Europe in general.
RESPONSE: The following sentences were added into the last paragraph (before description of the text structure): "The paper represents the first detailed and comprehensive analysis of weather-related fatalities in the Czech Republic, with particular respect to spatiotemporal variability and the basic features that underlie them. Its results may make a significant contribution to disaster-risk reduction in the Czech Republic. At the same time, it is an important addition to knowledge of weather-related fatalities at a central European scale; studies addressing this matter, apart from certain papers cited above, have been somewhat sparse to date."

*Further comments:*

L33: The authors mention world-wide or continental scales; but there are also many studies and articles that examine fatalities due to weather events or natural hazards at the national level. Is there a reason why these are not mentioned here?

RESPONSE: Yes, we are starting here with fatalities in world-wide and continental scales. Then in the second paragraph we move to Europe, including national levels, as well as in the 3rd and 4th paragraphs. The 5th paragraph concerns of the Czech Republic. It means, that papers related to national levels are represented here very well.

L34: Consider changing to "presented a broad world-wide" (instead of "broader")
RESPONSE: It should be understood in the context. The analysis mentioned in the previous sentence was done for 92 countries and Holle (2016) was working in a broader, world-wide scale. But we can delete it and change as follows: "Holle (2016) presented a world-wide overview of lightning fatalities."

L40-41: This is a little confusing. The authors write that "particular attention has been paid to deaths associated with heat-waves and floods". Where? In this article specifically or is this meant in general?
RESPONSE: There is written: "Europe also has a very serious problem; particular attention has been paid to deaths associated with heat-waves and floods on this subcontinent." We changed it "on this continent" to express clearly that it concerns Europe.

The focus within this introduction on heat waves and floods should be better described and justified. After all, almost half of the fatalities described in this study are due to traffic accidents in problematic weather situations (ice and snow covered roads, fog and rain). Why is this not addressed in the Introduction? Are there no compilations and studies on this kind of incidents in research?
RESPONSE: We believe, that presentation of papers dealing with heat waves and floods is described in sufficient way and need not to be further extended. Moreover, these topics, particularly related to our paper, are further discussed in a greater detail in Discussion. Concerning of traffic accidents in problematic weather situations, following paragraph was added to Introduction: "Inclement weather conditions, such as glaze ice, hoar-frost, snow, rain, fog, etc., may contribute to the occurrence of vehicle accidents accompanied by casualties. On a wider scale, there exist many papers that address the effects of various weather conditions and floods on vehicle transport and accidents (e.g. Andrey at al., 2003, 2010; Eisenberg and Warner, 2005; Brijs et al., 2008; Diakakis and Deligiannakis, 2013; Jackson and Sharif, 2016; Han and Sharif, 2020a, 2020b). However, studies of weather-related casualties arising out of vehicle accidents are absent for the Czech Republic. For example, without reference to fatalities, only case studies related to hoar-frost or glaze-ice situations (e.g. Sulan, 2006; Zahradníček et al., 2018), or damage to road network caused by natural disasters (Bíl et al., 2015), are available."
New references:
Andrey, J., Karlis, D., and Wets, G.: Long-term trends in weather-related crash risks, J. Transp. Geogr., 18, 247–258, https://doi.org/10.1016/j.jtrangeo.2009.05.002, 2010.
Andrey, J., Mills, B., Leahy, M., and Suggett, J.: Weather as a chronic hazard for road transportation in Canadian cities, Nat. Hazards, 28, 319–343, https://doi.org/10.1023/A:1022934225431, 2003.
Bíl, M., Vodák, R., Kubeček, J., Bílová, M., and Sedoník, J.: Evaluating road network damage caused by natural disasters in the Czech Republic between 1997 and 2010, Transp. Res. A, 80, 90–103, https://doi.org/10.1016/j.tra.2015.07.006, 2015.
Brijs, T., Karlis, D., and Wets, G.: Studying the effect of weather conditions on daily crash counts using a discrete time-series model, Accident Anal. Prev., 40, 1180–1190, https://doi.org/10.1016/j.aap.2008.01.001, 2008.
Diakakis, M. and Deligiannakis, G.: Vehicle-related flood fatalities in Greece, Environ. Hazards-UK, 12, 278–290, https://doi.org/10.1080/17477891.2013.832651, 2013.

Eisenberg, D. and Warner, K. E.: Effects of snowfalls on motor vehicle collisions, injuries, and fatalities, Am. J. Public Health, 95, 120–124, https://doi.org/10.2105/AJPH.2004.048926, 2005.
Han, Z. and Sharif, H. O.: Investigation of the relationship between rainfall and fatal crashes in Texas, 1994–2018, Sustainability–Basel, 12, 7976; https://doi.org/10.3390/su12197976, 2020a.
Han, Z. and Sharif, H. O.: Vehicle-related flood fatalities in Texas, 1959–2019, Water–Sui, 12, 2884, https://doi.org/10.3390/w12102884, 2020b.
Jackson, T. L., and Sharif, H. O.: Rainfall impacts on traffic safety: rain-related fatal crashes in Texas, Geomat. Nat. Haz. Risk, 7, 843–860, 10.1080/19475705.2014.984246, 2016.
Sulan, J.: Jíní – jev nebezpečný pro silniční dopravu (Hoar-frost as a danger phenomenon for the road traffic), Meteorol. Zpr., 59, 37–42, 2006.
Zahradníček, P., Münster, P., Bíl, M., Skalák, P., Panský, M., Brzezina, J., Bílová, M., and J. Kubeček: The December 2014 glaze event in the Czech Republic: predictability and impacts, Weather, 73, 375–382, https://doi:10.1002/wea.3199, 2018.

L53: Please reconsider the formulation "although other countries have their shares"
RESPONSE: Despite this English formulation is correct, the other formulation could be: "although similar studies exist also in other countries"

L54: The article by Hilker et al. (2009) seems to primarily focus on financial damage caused by floods, landslides and other processes. I guess it is by all means citable here but for Switzerland, Badoux et al. (2016) would probably fit a bit better because it focusses on natural hazard fatalities.
RESPONSE: In this paragraph we mention only papers related to flood- and landslide-related fatalities, i.e. we cited here Hilker et al. (2009) as example. Badoux et al. (2016) with focus on natural hazard fatalities is cited in the following paragraph.

Line 54&57: The article by Petrucci et al. you cite first in the course of the text should be labelled 2019a (and not 2019b)
RESPONSE: Attributing a or b to the year of citation does not reflect order in the text, but order following from alphabetical list of references. From this point of view, it is correct.

L62-63: The Salvador et al. (2020) study should rather be mentioned in the second paragraph of the Introduction where studies on heat waves and droughts are addressed. It there a special reason it is placed here?
RESPONSE: We placed it here in paragraph reporting more mixed national fatalities from other reasons (e.g. lightning, drought) than those that has been reported in the previous paragraphs (e.g. heat-wave, flood-landslides).

L74-75: When the authors mention "The general increase in climatic and weather hazards…" do they mean an increase in the number of events? Please include a literature citation that supports this statement.
RESPONSE: We changed the sentence as follows: "The general increase in frequency and severity of climatic and weather hazards (IPCC, 2012, 2013; Hoppe, 2016) …"
New references:
Hoppe, P.: Trends in weather related disasters – Consequences for insurers and society, Weather Clim. Extremes, 11, 70–79, https://doi.org/10.1016/j.wace.2015.10.002, 2016.
IPCC: Managing the Risks of Extreme Events and Disasters to Advance Climate Change Adaptation. A Special Report of Working Groups I and II of the Intergovernmental Panel on Climate Change, edited by: Field, C. B., Barros, V., Stocker, T. F., Qin, D., Dokken, D. J., Ebi, K. L., Mastrandrea, M. D., Mach, K. J., Plattner, G.-K., Allen, S.K., Tignor, M., and Midgley,

P. M., Cambridge University Press, Cambridge, United Kingdom and New York, USA, 582 pp., 2012.

IPCC: Climate Change 2013: The Physical Science Basis. Contribution of Working Group I to the Fifth Assessment Report of the Intergovernmental Panel on Climate Change, edited by: Stocker, T. F., Qin, D., Plattner, G.-K., Tignor, M., Allen, S. K., Boschung, J., Nauels, A., Xia, Y., Bex, V., and Midgley, P. M., Cambridge University Press, Cambridge, United Kingdom and New York, USA, 1535 pp., 2013.

L78: "The work is based on its own mortality database…" is a bit an awkward formulation. Consider making the text clearer, for example like this: "The work is based on the mortality database compiled by the authors from newspaper data and other official/administrative sources of information"
RESPONSE: Thanks for this proposal, we gladly accept this sentence.

L79-81: Consider changing to "The results in Section 4 describe weather-related fatalities for various weather phenomena and for all phenomena combined". Moreover, the second part of the sentence on line 80 does not match the first grammatically.
RESPONSE: Thanks for this proposal, we gladly accept this sentence.

**Data**

The three examples of fatality records (L99-116) could be moved to the appendix to shorten the text.
RESPONSE: Accepted and done as requested.

*Further comments:*

L99: Change to "One of the fatalities of the disastrous August 2002 flood was…"
RESPONSE: Thanks, changed as requested.

L133: What are "normal weather conditions"? This should be explained. I guess the authors mean dry weather with good visibility etc. I am not sure if "normal" is the good term, because would mean "rain" is generally an abnormal phenomenon (which is not true).
RESPONSE: We used terminology which is applied in police yearbooks without any further definitions/explanations. Generally, "normal weather conditions" mean everything what is not included into other categories mentioned. To avoid some doubts and misunderstanding, we changed the sentence as follows: "This includes the numbers of fatalities occurring during fog, the onset of rain …."

**Methods**

The typification of weather events to distinguish weather-related fatalities is not consistent. This is actually my most important criticism of this otherwise very carefully and precisely edited project/article. In my opinion, there is a big confusion between (1) weather phenomena, (2) resulting hydrological and geomorphological processes, (3) resulting traffic accidents, and (4) resulting other accidents.
RESPONSE: We added following explanations to the use of the term "weather-related fatalities" in Methods as a response to your comment: "For the purposes of this study, the general term "weather-related fatalities" refers to all fatalities directly attributable to meteorological or hydrological phenomena (windstorm, lightning, flash-flood, etc.), or those in which weather phenomena contributed to circumstances that finally led to death(s) in

combination with other factors (e.g. vehicle accidents during inclement weather conditions). Thus, this approach does not represent the occurrence of meteorological or hydrological extremes in the statistical sense based, for example, on return periods or low percentiles derived from corresponding statistical distributions. Because of the great variety of different weather-related effects, any reasonable presentation of fatality numbers and their basic features must be categorized; for the purposes of this paper, they have been divided into ten categories, which are described in Table 1." Including Table 1 instead of description of individual categories in the manuscript follows the request of the editor.

If floods are considered, why, for example, are mass movements such as landslides, rockfall, and debris flows not considered (at least not mentioned)? These can also be triggered by precipitation. The inconsistency of the present approach is also shown by the fact that avalanches (a mass movement) are taken into account within the weather type "snow", but the above-mentioned processes (various landslide types) do not seem to have been considered anywhere.

RESPONSE: Two detected fatalities caused by landslide were attributed to category of floods, during which they occurred. Other three landslide fatalities with other circumstances and without relation to flood were included into the category "other events" – see our response to L539.

Furthermore, considering road accidents in ice, snow, rain and fog is problematic in my opinion. I am familiar with e.g. the studies by Diakakis & Deligiannakis (2013, Vehicle-related flood fatalities in Greece) or by Coates (1999, Flood Fatalities in Australia, 1788-1996). Both publications show that people often die by drowning when they act carelessly in cars during floods and, for example, enter flood areas / plains or try to cross a watercourse during floods, be it over a bridge or by using a ford. These are flood victims in my opinion. You can also call them traffic victims, but they died mainly because of a natural hazard. But if a driver does not adapt his driving style to the external circumstances (on a wet road, snow-covered road, icy road or in poor visibility) and is involved in a crash, that is for me first and foremost a traffic accident. Of course, the "bad" weather plays a role and the victim can be classified as a weather-related fatality, but a clear distinction must be made from, for example, lightning fatalities, flood fatalities, etc., and the topic should be discussed in detail.

RESPONSE: Cases of fatalities reported on the example of Greece or Australia are generally not occurring in the Czech Republic (we have only one fatality when car was taken away from the road by torrent of water during flash flood and a driver drowned – it was included into category "flood". We understand your point of view and we speak about "vehicle accidents in relation to weather conditions" (L446). We agree with sentences you are writing about traffic victims and the role of "bad" weather and from this reasons we take these cases as "indirect" and "hazardous" cases. Moreover, we clearly separated these type of fatalities in Figure 11 and in accompanying text on L384-398. Further we changed slightly the sentence on L384 to be more clear as follows: "Because vehicle accidents are the cause of death for nearly half of the weather-related fatalities …"

*Further comments:*

L147: Regarding the "locality": I suppose this is the locality of the accident or event and not the actual place of death (which can also be in the hospital, for example). Perhaps this could be added.

RESPONSE: Exactly what you mention. We added it in brackets: "(ii) locality (i.e. place of the accident or event);"

L155-158: Why do the authors not use one of the three examples described in section 2.1 (L99-116); that would make sense in my opinion.

RESPONSE: We used another example, covering more-or-less all categories in database reported in the previous paragraph, what is not case of three examples from Section 2.1 with some missing features of fatalities. Moreover, they were moved now to Appendix.

L160: Maybe indicate that (in contrast to flash flood) these are triggered by long-lasting rainfall; thus consider "single-day or multi-day persistent rainfall" or something similar

RESPONSE: Our description is as follows: "Flood: This includes floods arising out of single-day or multi-day rainfall during precipitation-rich synoptic situations (rainy floods), of sudden melting of deep snow cover (snow floods) and of a combination of snow-melt and rainfall, sometimes with ice jams on the rivers (mixed floods) on the one hand, and flash floods arising from cloudbursts or torrential rains during thunderstorms on the other." We believe, that the whole context of this long sentence expresses all what the reviewer is saying.

L161: I would use the term "snow-melt flood" instead of "snow flood"

RESPONSE: We applied standardly used hydrological terminology in the Czech Republic with division to rainy, snow and mixed floods. Moreover, our formulation is: "of sudden melting of deep snow cover (snow floods)". We believe that this expression is clear enough.

L162: Change to "in the rivers"

RESPONSE: Thanks, changed as requested.

L162-163: Here is a typical case where it is difficult to distinguish the categories (see general comment on methodology): How are fatalities from the category "floods / flash floods" and "convective storm" distinguished? Both are triggered by very intense, short-term precipitation events. The former category is the consequence, the latter category is the cause. This does not work for me.

RESPONSE: What is included into ten categories follows from their description in Methods. In this particular case you mention it is expressed as follows (to downpours in category (iii) we added "not causing a flash flood" to make it more clear) – now see Table 1:

"(i) Flood: This includes floods arising out of single-day or multi-day rainfall during precipitation-rich synoptic situations (rainy floods), of sudden melting of deep snow cover (snow floods) and of a combination of snow-melt and rainfall, sometimes with ice jams on the rivers (mixed floods) on the one hand, and flash floods arising from cloudbursts or torrential rains during thunderstorms on the other.

(iii) Convective storm: This includes phenomena associated with the development of cumulonimbus cloud, such as very strong wind (e.g. squall, tornado, downburst), lightning strike, downpour (not causing a flash flood), and hail."

L164: To distinguish this category (ii) from the next (iii), I propose to call them "Non-convective windstorms" or "(Non-convective) windstorms". Maybe even: "Non- convective, non-tornadic windstorms"

RESPONSE: We believe that terms "windstorm" and "convective storm" clearly distinguish cases of strong winds. Moreover, we define cases of windstorms compared to convective storms clearly (now see Table 1): (ii) Windstorm: Strong winds resulting from large horizontal gradients of air pressure, lasting from a few hours to some days, are considered windstorms.

L167: Downpour (short and very intense rainfall) and hail are phenomena that potentially trigger overland flow and subsequent inundations as well as floods in water courses, overflow and subsequent inundations (in mountainous regions intense rainfall can also trigger e.g.

channelized debris-flows); as I understand it, these process chains are covered in (i). What I don't understand exactly is whether there were people in the Czech Republic who were killed directly by heavy precipitation (very unlikely) or hail (possible but unlikely).
RESPONSE: If any downpour led to flash flood, it was included in category "flood". Hail was not participating in any such flood event (only one fatality was related to traffic accident that happen on the road covered "by layer of hails"). As we reported above, some weather patterns/phenomena represent circumstances contributing finally to death. From this reason it is important to decide about "direct" and "indirect" fatalities, which is specified in the analysis of ten individual weather categories.

L168: Similar to the wind, I would call this category (if it remains so) "Non-convective rain".
RESPONSE: We believe, that in the name of category we need not to identify physical origin of phenomena (now see Table 1): "(iv) Rain: This includes, in particular, rain and wet street communications surfaces/tracks." Moreover, we have to be aware, that newspaper information does not allow in many cases to say, what kind of rain it was – see also Czech police database, just reporting phenomena like rain, snow, fog etc.

L168: Same here (as comment on line 167): were fatalities in this category or event/phenomena type killed by rain? I already mentioned it above in my general comments on the Methods (and will address it again below regarding lines 552-567): I am not sure whether a traffic accident that occurs during rainfall should (always or per definition) be included in the analysis of weather-related climate fatalities. If a person drowns in their car during a flood, it's a clear-cut case. But if a person does not maneuver properly in wet road conditions, it is not a clear case to me. After all, driving on wet roads is a very common thing. The actual cause of an accident can then also be a completely different one.
RESPONSE: Please see our new explanations to the term "weather-related fatalities" above.

**Results**

Section 4 (Results) is extremely detailed and thorough with its nine subsections presenting the various hazardous weather phenomena. The question arises whether all figures 1 to 9 belong in the actual article. Alternatively, Figure 10 (which covers all weather phenomena) could be printed on a full page. The points on the map could be colored according to their type (or different symbols could be used). For the interested reader, Figures 1 through 9 would then be placed in a supplementary material appendix. They contain a great deal of important information, but lengthen the text massively. However, this is the decision of the associated editor or the editor-in-chief. Both options are actually OK for me
RESPONSE: We do not agree that nine subchapters and related figures should be placed in a supplementary material. What will remain in that case from the article? Text in subchapters and related figures should be presented together. One of the aim of the article is to show fatalities particularly according to several weather-related categories. Printing Figure 10 in a full page is sure no problem. We considered expression of the map in this figure in colours, but finally it was impossible to express there all 10 categories in any reasonable form.

*Further comments:*

L191: I'm not sure if "events" is quite correct in this subtitle; perhaps "categories" or "types" would be more appropriate
RESPONSE: Thanks, we accept "categories".

L195: Change to "heavy flood in August"

Maybe the authors can make a reference to an article or report which describes the event?
RESPONSE: Flood in August 2002 was considered nearly a millennial flood in the Czech Republic. From this reason we prefer at least "an exceedingly heavy flood". As requested, we added new reference:
Hladný, J., Krátká, M., and Kašpárek, L. (Eds.): August 2002 catastrophic flood in the Czech Republic, Ministry of the Environment of the Czech Republic, Prague, Czech Republic, 44 pp., 2004.

L197: Change to "In terms of monthly distribution,"
RESPONSE: But it is distribution during the year, not during the months. From this reason we prefer our formulation. Please see also our response to L655.

L210: Instead of "type of death" I would rather write "type of fatality" here in the caption, exactly the same as on line 153. Please check the entire document and use consistent term.
RESPONSE: Thanks, we changed on "type of fatality" everywhere.

L210: Instead of "Symbols", consider using "Abbreviations"
RESPONSE: Thanks, we changed to "Symbols and abbreviations" everywhere. It is necessary due to the fact, that there is a mixture of symbols and abbreviations.

L237: As already mentioned at line 167, I do not doubt that hail can kill people, but this number (8) seems to be quite high.
In the meantime, however, I think I have understood: these are probably road accidents, right? If so, that would have to be indicated here absolutely, otherwise this leads to misunderstandings).
RESPONSE: Not any hail killed people. Number 8 is related to "downpour or hail" together, all during vehicle accidents.

L238: I suggest changing to something like "were simply indicated as having occurred "during a thunderstorm"
RESPONSE: Thanks, changed as requested.

L249-250: Change to "4 – rain or hail", same as on line 237-238 (please be consistent); also, "3 – thunderstorm" is confusing, change to something like "3 – unclear, occurred during thunderstorm"
RESPONSE: Changed as requested: on line 237-238 on "downpour or hail" and in figure caption on "3 – during a thunderstorm".

L253: As mentioned above (Methods) I would use "4.1.4 Non-convective rain" as a subtitle
RESPONSE: Please see our expression to your comments in Methods.

L256: Change to "In terms of monthly distribution,"
RESPONSE: Please see our expression to L197.

L259: It is not clear to me what the authors mean by "smaller regions" of the Czech Republic. Try to describe more clearly here.
RESPONSE: Our formulation is: "Rain-related fatalities were distributed over the whole Czech Republic, with a higher concentration in some of the smaller regions and lower frequency near borders, for example, north-western, south-western and southern Bohemia and south-western Moravia (Fig. 4c)." If you will look on Fig. 4c, you will see there a higher concentration of places in many smaller regions (areas, clusters). How to explain it better?

L260-261: "All these fatalities were classified as indirect consequences of vehicle accidents." I do not understand this statement. To me, these fatalities appear to be a direct consequence of a vehicle accident and an indirect consequence of a non-convective rain event.
RESPONSE: Following of your comment, we propose a new formulation of the sentence as follows: "Because all these fatalities occurred as consequences of vehicle accidents, they were classified as "indirect" with respect to accompanying inclement weather."

L275: With "smaller areas", do the authors mean "clusters"?
RESPONSE: Please see our comment to L259.

L294: Change to "and 9 % occurred on"
RESPONSE: Thanks, changed as requested.

Figure 10: Using a color code in the bar charts is actually a good idea, but some of the colored sections are just too small or narrow to be recognized. Maybe an alternative way of displaying the proportions of the different weather types could be found (and placed e.g. in the appendix).
RESPONSE: We are not sure, how important is to know the exact portions of each of individual 10 categories in all characteristics of parts a-b and d-j in Figure 10. If this figure will be printed on full page, as proposed, it should be better recognised. We are not sure that such details are important for readers and should be included in form of any large table into Appendix.

L381: I would delete "(56.1 % altogether)", it is not really necessary
RESPONSE: Thanks, deleted.

L382: "Nearly a third of them (30.3 %) fell victim to"; this is confusing. Do the authors mean nearly a third of the 66 or of the 13?
RESPONSE: Changed as: "Nearly a third of all non-Czechs (30.3 %) fell victim to frost".

L387-388: The end of this list is a bit confusing.
RESPONSE: We changed it to make it more clear as follows: "and other events – 16 (2.8 %), of which nine fatalities were generally associated with thunderstorms."

L393-394: I do not understand this sentence:
"Using the vehicle accident casualties classified within "indirect deaths" and "hazardous behaviour" (96.8 %), 94 % of them died on roads and the remaining 6 % in built-up areas and the countryside." Please try to clarify this.
RESPONSE: We use the new formulations as follows: "All fatal vehicle accident casualties were classified as "indirect deaths" and most of them (96.8%) fell within the "hazardous behaviour" category, while 94 % of all fatal casualties occurred on roads between towns or villages and the remaining 6 % in built-up areas and the countryside."

L405: Consider changing subtitle to: "4.3.1 Official demographic databases (CSO)"
RESPONSE: Because of use "official" in subtitle 4.3, we change it as "4.3.1 Demographic database of the CSO".

L444: Consider changing subtitle to: "4.3.2 Police database of vehicle accidents"
RESPONSE: Thanks, changed as requested.

L448: For clarity, consider extending to: "A mean of 879.4 total fatalities per year was recorded for 2000–2019 due to traffic accidents, of which"

RESPONSE: Thanks, changed as follows: "A mean of 879.4 fatalities per year due to vehicle accidents was recorded for 2000–2019, of which …".

L450: Is "deteriorating" really the right term here? In the caption of Fig. 13 you use "inclement". Consistency is important
RESPONSE: Thanks, changed on "inclement".

L459-462: Please state clearly in the caption which data is shown here (own database or Czech police database). This is especially important for those readers who first look at all the figures before (perhaps) reading text.
RESPONSE: Thanks, we complemented sentence: "Data according to police database of vehicle accidents"

Also, what is "normal" (weather); where is the threshold between normal and not normal (inclement); this needs to be addressed. I also mention this point in my comment to the discussion.
RESPONSE: Sorry, but we are not able to define what is "normal" in Czech police database and we believe, that it is not necessary here. Based on this database, there are simply mentioned only several events (which we included under term "inclement weather" in this article) like rain, snow, glaze ice, fog …

Figure 13: Which color shows "normal" weather in Figure 13? Have you declared that?
RESPONSE: To avoid problem of "normal" weather we changed caption of Figure 13 as follows: "Figure 13. Fluctuation in (a) the annual number of all vehicle-accident fatalities including those during inclement weather conditions and …" In this connection we prepared also the new version of Figure 13 with a changed colour in part (a).

[Figure]

**Discussion**

The **first subsection** of the discussion (5.1 Data uncertainty) is well structured and very detailed (perhaps almost a bit too long/detailed). It covers the important points that control data uncertainty. The part between lines 473 and 481 could possibly be streamlined a bit.

RESPONSE: We are reporting here different facts which could be responsible for differences in reporting weather-related fatalities. From your expression it is not fully clear in what directions or what should be "streamlined".

The reviewer would be very interested in one additional point. The study was made (mainly) by means of information from the print edition of one daily newspaper (Právo) and its Internet counterpart (Novinky.cz). The use of different editions devoting space to individual regions of the Czech Republic ensured that decentralized events could also be identified. But to what extent? I would like the authors' assessment of whether the study would have benefited from (i) examining several daily newspapers and/or (ii) including regional and local newspapers. It is clear to me that this endeavor would probably have been far too burdensome, but I think this question should be briefly discussed in this section of the manuscript.

RESPONSE: To obtain more-or-less homogeneous fatality datasets, we decided for one type of newspaper (Právo) and related internet part (Novinky.cz), where we used besides the "main issue" also all their regional issues. This was motivated also by our past and future research activities in this field, because Právo (Rudé Právo) was the main newspaper used in the Czech Republic and former Czechoslovakia before 2000. The greater part of information was taken from central "Czech Press Office", reports in other regional and local newspapers had the same source of information, sometimes complemented by local news. It is clear that what we have collected in our database represent lower estimate of real weather-related fatalities but we do not have any tool how to quantify it or be more correct to say where we really are and what portion of fatalities is missing, what is a usual feature of documentary evidence. Further improvement of our database is a problem of greater personal/financial possibilities what is not a question of weeks or months. Sorry, it is a real situation.

In the **second subsection** of the discussion (5.2 Weather-related fatalities in diff. databases) the authors mention the most important point at the beginning on line 505: the different data sources "are not (fully) comparable". The reviewer would even say they are not (really) comparable. (In all honesty,) This somewhat reduces my interest in this part of the study (or discussion). Nevertheless, it is of course important that the authors point out the problems with the comparative use of data from different sources. And they do it extensively and completely. If the editor concludes that the paper needs to be shortened somewhat, it could be amended here. Figure 14 and its description in the text, while interesting, takes up quite a bit of space. A more concise approach could certainly be taken here.

RESPONSE: Figure 14 we see as quite important. It gives some example allowing certain comparison of data coming from two databases (our and CSO) for similar reason of deaths, showing also with real meteorological patterns (characteristics). It is also example of quantification of relationship between fatalities and meteorological characteristics. We see it as an important part of discussion. It means, we would like to preserve this part of the paper.

The **third subchapter** of the discussion (5.3 The broader context) is, in my opinion, a bit long and, above all, not very well organized (with the exception of the very well worded and meaningful last paragraph, L 638-648). I do not find a common thread in it. I feel that the authors list a bit too many comparisons with various other studies (Czech an and international) in this subsection. The order in which they are mentioned does not always seem clear and partly I am not sure that all comparisons are very helpful in assessing the facts and figures in this article.

RESPONSE: We believe that comparison of results in this article with other existing studies is quite important giving well comparison with other regions and stages of similar research. We are trying to put our results into this broader context. From these points of view, we see it rather as a positive than a negative feature of this study.

*Further comments:*

L465: Avoid using "created" twice in the same sentence. Maybe change to: "The new database of weather-related fatalities developed for the purposes of this study…"
RESPONSE: Thanks, we changed as: "The newly-created database of weather-related fatalities developed for the purposes …".

L471-472: Consider changing to "over a 20-year period may". For, yes, societal changes took place during these 20 years, but this time period does not represent a very long time period compared to study periods in other similar investigations such as 200 plus years in Coates (1999) or 50 years in Atkins and Williams (2000).
RESPONSE: Thanks, changed as requested.

L506: The reference should be made to (Fig. 12, X31) and not to Fig. 11
RESPONSE: Thanks, corrected.

L510: In "These series included mean DJF temperature, mean minimum temperature Tmin,…" are mean *daily* temperatures and mean *daily* minimum temperatures meant?
RESPONSE: We suppose that there is climatologically clear that all temperature characteristics reported were calculated from daily values. We corrected the following sentence as follows: "These DJF series included mean temperature, …"

L531: consider extending to "23 casualties against 15 in our data collection."
RESPONSE: Thanks, corrected.

L533-534: I do not know the spatial distribution and density of meteorological stations in the Czech Republic (the values are probably comparable with the values in other European countries), but since convective thunderstorms can occur on a very small scale, this argument is not necessarily a strong one.
RESPONSE: The station network of the Czech Hydrometeorological Institute includes *c*. 250 climatological and *c*. 1000 rain-gauge stations. On all of them meteorological phenomena (like thunderstorms) are recorded. It means that it is quite dense network of stations, generally decreasing probability that any thunderstorm would be not recorded anywhere. On the other hand, we only mentioned the fact, that no thunderstorms were reported in the corresponding day on those stations – we are not saying, where is a problem.

L534-535: I do not understand "From the other point of view". Why "other", I am a bit confused. The 26.02.2003 and 14.10.2017 incidents come from the CSO database, right?
RESPONSE: Thanks, formulation "From the other point of view" was deleted.

L539: This is the first time in the text that landslides are mentioned. Up to this point, it is not clear to the reader that landslides are considered in this compilation. This would have to be adjusted in chapter 3.
RESPONSE: We complemented to Section 3 the new category: "Other events: cases of very rare events that could not be attributed to any of the previous categories (landslides, rime, or simply "bad" weather)." In fact, this category includes only 7 fatalities, from which 3 casualties

were caused by landslides, 1 due to rime on trees and 3 during generally "bad weather". We used this category from reason of complexity only in Figure 10.

552-567: The statements made by the authors in this paragraph confirm my concern and reluctance to include (all) traffic accidents in the compilation of an overview of weather- related fatalities. This is because, in a sense, it is a question of "thresholds." Take, for example, the category "Rain" in the present database, or "Rain" and "Onset of rain and light rain" in the police data. With an annual precipitation total of 450 to 550 mm in the Prague region, a light precipitation of 5 mm in say 4 hours cannot be considered as "abnormal" weather (there are probably about 80-100 rainy days in Prague annually).  A traffic accident occurring in such conditions cannot, in my opinion, result in a weather- related fatality.
RESPONSE: Please see our earlier explanations. Weather factors represent important "circumstances" contributing to vehicle accidents in connections with driver's behaviour. For example, if there was any rain, the road was wet and became slippery. We do not speak about "abnormal" weather as explained earlier. We believe that such data should be included to our analysis.

Theoretically, only cases that occur as a result of exceptional weather should be taken into account. Or how do the authors see it? I think this needs to be addressed in this article. Even more so when you consider that 563 of the 1164 deaths (48.4%) in this database are traffic accidents. I dare to say that this percentage is quite extraordinary compared to countries worldwide (I don't think the authors mention this in section 5.3, for example).
RESPONSE: Sorry, but our opinion is different. There is not clear, how to evaluate "exceptional weather". If the road is slippery being wet, with snow or glaze ice, how to evaluate what is exceptional? There is not information, if there was 1, 5 or 10 cm of snow on the road – simply it was slippery due to snow. Reports of accident do not go after such details. Information about 48.4% is included in L370. We do not know similar papers to estimate if "this percentage is quite extraordinary compared to countries worldwide".

L586-589: I do not think these two studies are comparable. The present work lists weather-related fatalities. The Swiss study, on the other hand, describes natural hazard processes and does e.g. not include frost and heat deaths, nor does it include traffic accidents that occur during "non-normal" weather conditions.
RESPONSE: Sorry, but why we cannot give example and results of studies dealing with fatalities in other parts of (central) Europe? We are not saying it is more/less comparable to the Czech Republic, we are just reporting their results showing situation in different natural conditions (the Alps and snow avalanches).

L590: Typo, please change to "an average of 42.6 fatalities a year;" (without closing parenthesis)
RESPONSE: Thanks, corrected.

L597: What does "their" (in "the structure of their total of") refer to? The entire EUFF or the "remaining regions"? Please try to be accurate here.
RESPONSE: Thanks, corrected as: "In more detail, the structure of total 2466 flood fatalities in the EUFF database detected features …".

L603-608: Since the article is already quite long and this type of accident occurs only slightly in the previous study, I would omit this section. The number 4000 from Sharma et al (2020) is actually worthless if no time period can be associated with it.

RESPONSE: We see this study as quite important, being one of those dealing particularly with this topic. It documents clear effects of warmer winters on this kind of fatalities, what can be important also for the future. Because the same type of fatalities we analysed also in our study, there is worthy to cite this research. For these reasons we would like to preserve this paragraph, where we added time period of the study: "… covering 10 Northern Hemisphere countries in the 1991–2017 period."

L611: Consider clarifying and changing to: "glaze ice, snow and for all weather-related fatalities. Windstorms show a significant decreasing linear trend only according to the Mann-Kendall test. The remaining types of weather-related fatalities returned statistically insignificant trends."
RESPONSE: Our formulation is: "… statistically significant falling linear trends were revealed by both methods of linear trend calculation for convective storms, glaze ice, snow and all weather-related fatalities, as well as for windstorms according to the Mann-Kendall test. The remaining groups of events returned statistically insignificant trends." We think that the first sentence is correct. The second sentence we can change according to your proposal as follows: "The remaining categories of weather-related fatalities returned statistically insignificant trends."

L619: I don't understand exactly what the authors mean by "total natural deaths." That has nothing to do with weather-related deaths, does it? If so, why is that of interest here?
I propose to significantly shorten the text from line 617 to 624.
RESPONSE: This is some misunderstanding. The term natural is here not used in sense "nature", but as "normal, usual" mortality (the term is used by authors of the cited paper).

L629-637: It should be taken into account that, in general, the number of traffic fatalities is decreasing in many European countries, regardless of the weather. This development is very clear, e.g. in Germany and France, where the number of traffic fatalities has decreased 5 to 6 times since the early 1970s. Many of the reasons for this are given by the authors in the text (L 633-635). The citation of the Audrey (2010) study is interesting, but this general trend should definitely be stated in this paragraph, for example at the end of it.
RESPONSE: Following text was added in this paragraph: "Similarly, a decrease in total and rain-related fatal crashes was reported for Texas (USA) in the 1994–2018 period (Han and Sharif, 2020a). Decreases in weather-related and total fatalities in vehicle accidents are also clearly expressed in European trends. For example, the number of fatalities in traffic accidents in Germany fell from 7503 in 2000 to 3046 in 2019 (BASt, 2021). Similar tendencies follow from road-death statistics across 31 European countries during the 2001–2019 period. Of these, 21 countries recorded relatively larger decreases than the Czech Republic and 9 countries relatively smaller (see ETSC, 2021)."
New references:
BASt – Bundesanstalt für Straßenwesen: Traffic and Accident data, Summary Statistics – Germany, available at: https://www.bast.de/BASt_2017/EN/Publications/Media/Traffic-and-Accident-Data.pdf?__blob=publicationFile&v=7, last access: 10 March 2021.
ETSC – European Transport Safety Council: Road deaths in the European Union – latest data, available at: https://etsc.eu/euroadsafetydata/, last access: 10 March 2021.

L631-632: I do not understand the use of "Because" at the beginning of the sentence. I suggest rephrasing (maybe making two sentences). The second sentence could start with "However, the influence..."
RESPONSE: Thanks, instead of "Because" we use "Although".

**Conclusions**

The conclusions are concise, which I like, and present the key findings of the study. They can be read and understood independently of the overall text, which is becoming increasingly important for the large number of so-called "cross-readers" these days.

I would add in point (iv), (v), or directly below these points that comparing fatality data from different sources within the same country is problematic or difficult (as shown in Sections 5.2 and 5.3) and that comparison with similar studies from other countries is also challenging (as shown in Section 5.3).

RESPONSE: We added as a new and final point: "(vii) A comparison of the numbers and features of weather-related fatalities from three different databases demonstrates the complexity and difficulty of analysis even within one country. Logically enough, comparison of such results with similar studies from other countries also presents a formidable challenge."

*Further comments:*

L651: Consider changing to "for the Czech Republic in the period 2000–2019,"
RESPONSE: Our formulation (in the 2000–2019 period) is OK.

L653: Please check "constitutes a unique data source for.. …structures of such casualties". What do the authors mean exactly by structures of such casualties? Clarify.
RESPONSE: Thanks, we used "features" instead of "structures".

L655: Change to "In the monthly distribution of fatalities,"
RESPONSE: We use the term "annual distribution" in the sense of "distribution of fatalities during the year". We have never been criticized for the use of this term in many our climatological studies or in our publication dealing with fatalities (Brázdil et al., 2019b). But we know that, for example, in the paper by Petrucci et al. (2019a) was preferred the term "monthly distribution". It is probably a question of subjective understanding of sense of this term.

L655-656: Consider changing to "The decreasing linear trends in fatalities from 2000-2019 were statistically significant only for fatalities caused by all weather factors combined and for fatalities caused by…"
RESPONSE: Thanks, corrected as: "The decreasing linear trends in the number of fatalities during 2000–2019 were statistically significant only for fatalities related to all weather factors together and …"

L659: Again, I have a Problem with the term "structure of fatalities"; but maybe the problem is with me. Do the authors mean (?):
"The composition of weather-related fatalities with respect to different distinguishing criteria indicates…"
RESPONSE: We used the term from the paper by Petrucci et al. (2019a) and changed a corresponding sentence as: "The basic features of weather-related fatalities indicates …"

**RESPONSES TO REFEREE 2**

**Comment on nhess-2021-14**
Anonymous Referee #2
* * *
Referee comment on "Fatalities associated with the weather in the Czech Republic, 2000–2019" by Rudolf Brázdil et al., Nat. Hazards Earth Syst. Sci. Discuss., https://doi.org/10.5194/nhess-2021-14-RC2, 2021
* * *
Very important paper of sophisticated quality. Both the methodology and the literature background are valuable. Although, the paper is rather long, 47 pages, the description of the methods and results are consequent and systematic, so I could not recommend any substantial abbreviation. The Figures are also informative and well edited.
RESPONSE: Thank you for general evaluation of our paper. Because this study represents the first basic paper with attention to fatalities related to weather in the Czech Republic in this complex view, we see as important to explain different aspects of this problem which is a reason that the paper is "rather long".

On p. 6, lines 164-167 you explain the difference between „windstorm" and „convective storm". Were these details all sufficiently included in the documentary sources, or you applied additional meteorological information to select?
RESPONSE: Because related newspaper reports described in detail also accompanying situation/patterns/phenomena, there was relatively very simple to distinguish between these two categories and there was not necessary to look on any additional meteorological information (e.g. from meteorological observations on any near station for any particular event).

On p. 6, lines 175-181 you describe the two significance tests applied. It is not clear however, when you included the regression line into the corresponding diagrams. If both tests demonstrated significance? Please clarify it, or, even better if you decide and process according to the Mann-Kendall test, only!
RESPONSE: We preferred calculation of linear trends based on the method of linear regression, including evaluation of their significance, which appears also in corresponding diagrams. The Mann-Kendall (M-K) test was used as a further tool to show if its application will give different information of trend significance. As follows from our explanations on lines 609-612, the M-K test identified additionally only trend in windstorm-induced fatalities as statistically significant.

Concerning the documentary sources, it would be useful to read the authors opinion and direct analyses on representativity of the long-term trend, annual cycle and distribution among the meteorological reasons established from the authors' data base. The requested analyses could compare the above three aspects derived from the documentary and official data sources.
RESPONSE: Concerning of long-term trends in our fatality database, statistically significant trends appear only in categories convective storms, glaze ice and snow, while in other categories they are insignificant. Because glaze-ice- and snow-induced fatalities result generally from vehicle accidents, these tendencies agree with general decreasing trends in casualties during such accidents and in increasing temperatures during the winter (winter half-year) contributing to less frequent occurrence of glaze-ice and snow. Because fatalities in vehicle-accidents represent nearly half of all weather-related fatalities in the Czech Republic, it is reflected also in their significant decreasing trends. Annual cycle of fatalities with a maximum

in winter months reflects well the occurrence of the most frequent categories – frost, glaze ice and snow. The secondary maximum in summer is attributed to floods, convective storms and rain. I.e., it reflects well also distribution of these hydrometerological phenomena during the year. Despite a dominant number of fatalities during vehicle accidents (induced by glaze ice, snow and rain), frost-induced fatalities are most frequent among other categories, followed by floods, which are generally taken as the most damaging and deadly natural disaster in the Czech Republic. I.e., we believe, that the analysis of fatalities based on our documentary database reflects reality representatively. On the other hand, as reported in the paper, our database deeply undervalues fatalities related to heat-waves which requires different type of data for the analysis. Section 5.2 shows comparison of our database with those of CSO and police, where different aspects of long-term trend, annual cycle and distribution among the meteorological reasons are discussed in detail needed.

**RESPONSES TO EDITOR**

Dear Prof. Brázdil and co-Authors,

Thank you for the submission of your very interesting manuscript "Fatalities associated with the weather in the Czech Republic, 2000–2019".

As you know, two reviewers have now provided detailed reviews, which you have replied in thoughtful detail to. Both reviewers recommended some revisions, and therefore I would like to invite you to submit a revised version of your manuscript.

Would you please also provide an 'author's reply' to the reviewers (feel free to use the same words that you used in what you have already uploaded). Please, also include a track changes document between the old manuscript and the new one (you can include this as part of your 'author's reply').

I read carefully the suggestion supplied by the referees and I think that their advices will improve the quality of your paper, giving more emphasis to the huge amount of work that you have done. In addition, I would like to suggest some general items.

I agree with R1 about the title because it seems not explanatory enough. One of the titles suggested by R1 for me should be good; alternatively, I suggest you "weather conditions".
RESPONSE: Accepted, we changed a title as follows: "Fatalities associated with the severe weather conditions in the Czech Republic, 2000–2019"

Both R1 and R2 noted that the paper is long, and maybe some efforts could be done to try to reduce it. I suggested the following modifications:
1. Section 2.1: I suggest to describe less in detail documentary sources, you can report their importance simply quoting some reference and then briefly describe the sources used in this work. I suggest to reduce the examples of descriptions because the research in documentary sources is not the focus of the paper: it is widely accepted in literature as a research tool, moreover these aspects are also clarified in discussion. This section should have similar length of 2.2 and 2.3.
RESPONSE: Accepted, we reduced the text in Section 2.1 on the length comparable to 2.2 and 2.3. On the other hand, we are not sure that potential readers will be familiar with this special type of data as documentary data represents. From this reason we moved three concrete examples into Appendix.

2. Lines from 160 to 174 could be easily changed in a 2-column table (Phenomenon/description) easier to find in case the reader have some doubts.
RESPONSE: Accepted, instead of text we used the new Table 1.

**Table 1. Categories of weather-related fatalities and their short description**

| Category | Description |
| --- | --- |
| flood | cases arising out of single-day or multi-day rainfall during precipitation-rich synoptic situations (rainy floods), of sudden melting of deep snow cover (snow floods) and of a combination of snow-melt and rainfall, sometimes with ice jams in the rivers (mixed floods); flash floods arising from cloudbursts or torrential rains during thunderstorms |
| windstorm | strong winds resulting from large horizontal gradients of air pressure, lasting from a few hours to some days |
| convective storm | phenomena associated with the development of cumulonimbus cloud: very strong wind (e.g. squall, tornado, downburst), lightning strike, downpour (not causing a flash flood), and hail |
| rain | rain and wet street communications surfaces/tracks |

| | |
|---|---|
| snow | snow calamity and avalanche |
| glaze ice | ice-patches or glaze-ice cover on streets, roads and communications |
| frost | severe frosts occurring as a part of cold spells, bodies of water insufficiently frozen for the activity undertaken on ice |
| heat | extremes of high temperature occurring in the course of heat waves |
| fog | cases of significantly decreased horizontal visibility due to fog |
| other events | cases of very rare events that could not be attributed to any of the previous categories (landslide, rime, "bad" weather) |

3. Figures from 1 to 9 could be changed, i.e., I suggest to reduce the size of the map that can be put top right in a square, near yearly and seasonal distribution. These maps should be giving simply an idea because, if these data must be used in a national project, the geocoded database must be used.
RESPONSE: We tried to follow this suggestion and put maps in top right. As we tested it for two figures 3 and 7, the maps were too small to be readable with a loss of necessary information, included in them. From this reasons we would be happy to preserve Figures 1-9 in the form, in which they were prepared originally.

4. Pag. 30 could be changed in a table, because for each phenomenon, the same group of parameters are supplied. I suggest something like this but you could improve it.
RESPONSE: Accepted, instead of text we used the new Table 2.

Table 2. Number of fatalities and their features according to the Czech Statistical Office database in the 2000–2019 period: W00 – fall on ice or snow, X30+X32 – excessive natural heat and solar radiation, X31 – excessive natural cold, X33 – lightning strike, X36+X37+X38+X39 – avalanche, landslide or other earth movements, natural catastrophic storm, flood (inundation), and other and non-specified natural forces

| Category | Number of fatalities | | | Sex (%) | | Age - year (%) | | |
|---|---|---|---|---|---|---|---|---|
| | total | max/year | month/% | male | female | 0–15 | 16–65 | >65 |
| W00 | 46 | 5/2003, 2008, 2010 | Jan/32.6 | 73.9 | 26.1 | 0.0 | 19.6 | 80.4 |
| X30+X32 | 38 | 7/2017 | Jun/34.2 | 71.1 | 28.9 | 2.7 | 52.6 | 44.7 |
| X31 | 2407 | 186/2010 | Jan/25.1 | 75.5 | 24.5 | 0.1 | 67.4 | 32.5 |
| X33 | 23 | 5/2008 | Aug/30.4 | 78.3 | 21.7 | 8.7 | 91.3 | 0.0 |
| X36+X37+X38+X39 | 31 | 7/2009 | Jun/25.8 | 83.9 | 16.1 | 9.7 | 77.4 | 12.9 |

Because by deleting the related text, replaced by Table 2, in this chapter would be only the first sentence and no other text, we added short characteristics of main findings: "Excessive natural cold (X31) was responsible for the highest number of weather-related fatalities: 2407 victims in the 2000–2019 period (Table 2). Fatalities of this type exhibited a statistically significant increasing linear trend, in similar fashion to fatalities arising out of excessive natural heat and solar radiation together (X30+X32) (Fig. 12). The percentages of male victims (71.1%–83.9%) predominated over those of females in each of the five fatality types (Table 2). While the elderly predominated (80.4%) in fatalities caused by falls on ice or snow (W00), adult deaths were the most frequent in the remaining four types. Higher percentages of elderly fatalities were also recorded for types X30+X32 and X31."

---

## Editor Decision (ED1)

*Dear Prof. Brázdil and co-Authors,*

Thank you for the submission of your very interesting manuscript "Fatalities associated with the weather in the Czech Republic, 2000–2019".

As you know, two reviewers have now provided detailed reviews, which you have replied in thoughtful detail to. Both reviewers recommended some revisions, and therefore I would like to invite you to submit a revised version of your manuscript.

Would you please also provide an 'author's reply' to the reviewers (feel free to use the same words that you used in what you have already uploaded). Please, also include a track changes document between the old manuscript and the new one (you can include this as part of your 'author's reply').

I read carefully the suggestion supplied by the referees and I think that their advices will improve the quality of your paper, giving more emphasis to the huge amount of work that you have done. In addition, I would like to suggest some general items.

I agree with R1 about the title because it seems not explanatory enough. One of the titles suggested by R1 for me should be good; alternatively, I suggest you "weather conditions".

Both R1 and R2 noted that the paper is long, and maybe some efforts could be done to try to reduce it. I suggested the following modifications:

1. Section 2.1: I suggest to describe less in detail documentary sources, you can report their importance simply quoting some reference and then briefly describe the sources used in this work. I suggest to reduce the examples of descriptions because the research in documentary sources is not the focus of the paper: it is widely accepted in literature as a research tool, moreover these aspects are also clarified in discussion. This section should have similar length of 2.2 and 2.3.
2. Lines from 160 to 174 could be easily changed in a 2-column table (Phenomenon/description) easier to find in case the reader have some doubts.
3. Figures from 1 to 9 could be changed, i.e., I suggest to reduce the size of the map that can be put top right in a square, near yearly and seasonal distribution. These maps should be giving simply an idea because, if these data must be used in a national project, the geocoded database must be used.
4. Pag. 30 could be changed in a table, because for each phenomenon, the same group of parameters are supplied. I suggest something like this but you could improve it

| | #F | #F/y | Max/Y | Years | Month/Max | Sex/max | Sex/Max/Age |
|---|---|---|---|---|---|---|---|
| Falls on ice or snow | 46 | 2.3 | 5 | 2003 2008 2010 | Jan/84.8% | Males/73.9% | Males/80.4%/Elderly |
| Excessive natural heat and solar radiation | | | | | | | |
| Others…. | | | | | | | |

I look forward to seeing the next version of your manuscript which I will not send out for further review, but rather, will make the decision myself, assuming no major items come up in the revised manuscript for which I need outside reviewers to aid me in my decision.

Kind regards,
Olga Petrucci
NHESS Editor